# High resolution satellite products improve hydrological modeling in northern Italy

Lorenzo Alfieri[1], Francesco Avanzi[1], Fabio Delogu[1], Simone Gabellani[1], Giulia Bruno[1], Lorenzo Campo[1], Andrea Libertino[1], Christian Massari[2], Angelica Tarpanelli[2], Dominik Rains[3], Diego G. Miralles[3], Raphael Quast[4], Mariette Vreugdenhil[4], Huan Wu[5,6], Luca Brocca[2]

[1]CIMA Research Foundation, University Campus of Savona, Savona, 17100, Italy
[2]National Research Council, Research Institute for Geo-Hydrological Protection, Perugia, 06128, Italy
[3]Hydro-Climate Extremes Lab (H-CEL), Ghent University, Gent, 9000, Belgium
[4]Department of Geodesy and Geoinformation, TU Wien, Vienna, 1040, Austria
[5]Southern Marine Science and Engineering Laboratory (Zhuhai), Zhuhai, 519082, China
[6]School of Atmospheric Sciences, Sun Yat-sen University, Guangzhou, 510275, China

*Correspondence to*: Lorenzo Alfieri (Lorenzo.Alfieri@cimafoundation.org)

**Abstract.** Satellite Earth observations (EO) are an accurate and reliable data source for atmospheric and environmental science. Their increasing spatial and temporal resolutions, as well as the seamless availability over ungauged regions, make them appealing for hydrological modeling. This work shows recent advances in the use of high resolution satellite-based EO data in hydrological modelling. In a set of six experiments, the distributed hydrological model Continuum is set up for the Po River Basin (Italy) and forced, in turn, by satellite precipitation and evaporation, while satellite-derived soil moisture and snow depths are ingested into the model structure through a data-assimilation scheme. Further, satellite-based estimates of precipitation, evaporation and river discharge are used for hydrological model calibration, and results are compared with those based on ground observations. Despite the high density of conventional ground measurements and the strong human influence in the focus region, all satellite products show strong potential for operational hydrological applications, with skilful estimates of river discharge throughout the model domain. Satellite-based evaporation and snow depths marginally improve (by 2% and 4%) the mean Kling-Gupta efficiency ($KGE$) at 27 river gauges, compared to a baseline simulation ($KGE_{mean}$=0.51) forced by high-quality conventional data. Precipitation has the largest impact on the model output, though the satellite dataset on average shows poorer skills compared to conventional data. Interestingly, a model calibration heavily relying on satellite data, as opposed to conventional data, provides a skilful reconstruction of river discharges, paving the way to fully satellite-driven hydrological applications.

## 1 Introduction

Remote sensing of the Earth from space is a ripe yet ever growing sector, with countless applications and users worldwide. Hydrological sciences have already benefited enormously from Earth observation data (see e.g., McCabe et al., 2017; Chen and Wang, 2018; Alfieri et al., 2018), thanks to global and independent datasets of the different components of the water and

energy cycles as well as anthropogenic processes such as irrigation (Massari et al., 2021). Hydrological models play a crucial role for monitoring and forecasting, thanks to their ability to reproduce the physical processes governing the water cycle. Their successful implementation is strongly conditioned by the availability of consistent, accurate, and seamless hydro-meteorological datasets for the considered focus region, space/time resolution and period of interest. Conventional data including ground observations and weather radars are traditionally favourite sources of dynamic data to force these models. Yet, they are not viable options for the still vast ungauged regions of the world. Satellite products offer a range of alternatives to fill such gaps, thanks to their massive contribution to the atmospheric reanalyses (see e.g., Hersbach et al., 2020) as well as with independent products. Hydrological models can benefit from dynamic data (either ground or satellite-based) in various forms: 1) as forcing datasets, 2) as assimilation datasets, 3) as benchmark data for model calibration and improved parameterization, and 4) to investigate process understanding.

Forcing data are mandatory input for hydrological models. Key variables are precipitation, air temperature and evaporation or, alternatively, the meteorological variables needed to estimate them. Their influence in hydrological modeling was assessed, for instance, by Wu et al. (2017) and Beck et al. (2017) for precipitation datasets, Dembélé et al. (2020a) for evaporation datasets, and Dembélé et al. (2020b) for combinations of temperature and precipitation datasets. The latter found a reduced influence of the choice of temperature datasets on the output discharge, though these can significantly impact evaporation and soil moisture estimates. Data assimilation methods are designed to merge measurements of any type with estimates from geophysical models (Reichle, 2008), to compensate for errors in the forcing data, model structural deficiencies, and update their state variables at the initial or intermediate simulation steps (Spaaks and Bouten, 2013). Relevant applications of assimilating satellite products in hydrological modeling include soil moisture (Massari et al., 2015; Wanders et al., 2014), water storage (Li et al., 2012), snow cover (Thirel et al., 2013), evaporation (Hartanto et al., 2017), land surface temperature (Silvestro et al., 2013), water levels (Paiva et al., 2013), discharge (Ishitsuka et al., 2021), water extent (Revilla-Romero et al., 2016; Hostache et al., 2018), and multi variable combinations (Wongchuig-Correa et al., 2020). Hydro-meteorological data has also been used as a benchmark to train the model parameters through machine learning techniques (Mosaffa et al., 2022) or calibration techniques based on minimization of cost functions computed between simulated and observed variables (Pechlivanidis et al., 2011, Demirel et al., 2018). Satellite estimation of river levels also shows promising applications in the field. It has been tested in the calibration of hydrological (Getirana et al., 2013; Dhote et al., 2021) and hydraulic (Domeneghetti et al., 2021) models.

As part of the Green Deal and the Digital Strategy, the European Commission recently launched the Destination Earth program[1], a joint effort involving key European institutions to develop a very high precision digital model, or "Digital Twin", of the Earth to monitor and predict environmental change and human impacts, to ultimately support sustainable development. The present work strives in that direction, by contributing to the development of a Digital Twin Earth focused on the water cycle and hydrological processes. It highlights the potential of high resolution satellite products in describing

---

1    https://digital-strategy.ec.europa.eu/en/policies/destination-earth

the water cycle and monitoring hydrological extremes and water resources. Through various dedicated experiments, we test the influence of five new high resolution satellite-derived datasets on the performance of CIMA's distributed hydrological model Continuum (Silvestro et al., 2013) set up for the entire Po River Basin in northern Italy. These include (1) GPM-SM2RAIN (Massari et al., 2020) precipitation and (2) GLEAM (Miralles et al., 2011) evaporation as dynamic forcing; data assimilation of (3) C-SNOW (Lievens et al., 2019) snow depth and (4) RT1 (Quast et al., 2019) soil moisture; and model calibration using (5) satellite-based river discharge (Tarpanelli et al., 2020) as a benchmark. By comparing results with observed river discharge over 2017-2019 and with a simulation forced by conventional data, we investigate the relative impact of these high resolution satellite products. Further, we take the first steps towards hydrological modelling fully relying on satellite data, by calibrating and subsequently running the model using SM2RAIN satellite precipitation and GLEAM evaporation as forcing, and satellite-based estimates of river discharge as benchmark data for the calibration.

## 2 Case study and data

### 2.1 Case study – the Po River basin

The Po River basin has a catchment area of about 74,000 km$^2$ shared between Italy (95%) and Switzerland (5%). It is fed by tributaries from the Alps in the North and West, and by the Apennines in the South. The basin elevation ranges between 4800m to the sea level, hence it features a variety of climatic and hydrological regimes, from glacial and snow-rain type in the mountain area, to a pluvial yet drier regime in the lowland section. The region is considered highly vulnerable to flooding, both economically and with respect to loss-of-life (Domeneghetti et al., 2015). The basin plays a significant role in the Italian economy, hosting approximately 25% of the italian population, producing 40% of the national GDP, and consuming 48% of national produced energy. The Po River flows through the Po Plain, one of the largest contiguous agricultural areas of Europe. This causes more than 30% of water to be extracted from surface water and used for agricultural purposes. Although water is sufficient for all uses under average climate conditions, recent periods of prolonged drought led to substantial economic losses and threats to water security (Mysiak et al., 2013), thus a comprehensive evaluation of the impacts of human activities on water resources in the area is a far-reaching matter. Given its large socio-economic influence, the Po basin has already been investigated through a number of modeling approaches forced by in situ data and by Earth system models, especially to predict the impact of inundation and of climate change (e.g., Ravazzani et al., 2015; Vezzoli et al., 2015; Nogherotto et al., 2019) while applications including satellite products remain scarce.

### 2.2 Static data

In the choice of spatial information, large scale datasets were deliberately used over more detailed local data, in line with the concept of the Digital Twin Earth and in view of the plan to extend the simulation area for a continental or global application. We used the Digital Elevation Model (DEM) from the global USGS Hydrologic Derivatives for Modeling and Analysis (HDMA, Verdin, 2017) at 3 arc-second spatial resolution (about 90 m at the equator), which comes with pre-

computed and corrected hydrological derivatives including channel network and macro basins. The DEM was upscaled at the chosen model resolution of 1 km through cubic resampling, to define the computational grid and compute the necessary hydrological derivatives (flow accumulation, drainage directions and channel network). The river network is defined by cells with an upstream area larger than 240 km$^2$, following previous applications of Continuum in northern Italy. To improve its spatial representation, the DEM was carved with a high resolution stream network of the main rivers taken from the Italian Institute for Environmental Protection and Research, while dikes were manually placed at specific locations, especially in flat areas.

The Curve Number map used to model direct runoff and infiltration from rainfall excess, was derived from the ESA-CCI 2018 Land Cover map (ESA, 2017) at 300 m resolution, together with information on the soil characteristics. Hydrologic soil groups were extracted from the HYSOGs250m (Ross et al., 2018), while for soil texture identification, we applied the USDA method (Shirazi and Boersma, 1984) using the ISRIC SoilGrids (Hengl et al., 2017) global maps of the fractions of sand and clay, combined with the ESA CCI SoilMoisture (Dorigo et al., 2017) global map of soil porosity. Glacier areas used in the cryospheric model S3M (see Section 3.1) were taken from the Randolph Glacier Inventory (RGI) v6 (Raup et al., 2007). Vegetation coverage is taken from the global land cover map ECOCLIMAP (Faroux et al., 2013).

Point information for a set of 99 reservoirs and the three major lakes (Maggiore, Como and Garda) was included in the model setup (Figure 1). Information on the dams and the corresponding reservoirs was provided by the Italian Civil Protection Department (DPC) and from the GranD database (Lehner et al., 2011). Data ingested for each dam include the maximum stored volume, initial volume, maximum non-damaging discharge at the outflow gates, weir length, maximum storage level, outflow coefficient, and coordinates of the release point. For lakes, required metadata are the outlet coordinates, minimum volume inducing outflow discharge, initial volume, and emptying coefficient.

## 2.3 Dynamic data

The hydrological model used requires input maps of precipitation, air temperature, relative humidity, wind speed and incoming solar radiation. Alternatively, both actual and potential evaporation can be provided as dynamic input, where the latter is used to estimate actual evaporation from lakes and reservoirs. In such case wind speed maps are not needed by the model. The baseline hydrological simulation uses conventional meteorological data as input. Precipitation fields were estimated with the Modified Conditional Merging (MCM) technique (Bruno et al., 2021), which incorporates precipitation gauges and radar estimates. MCM is an improvement of the Conditional Merging proposed by Sinclair and Pegram (2005), which estimates the structure of covariance and the length of spatial correlation at every gauge, taking it from the cumulated radar precipitation fields. For the Po River basin, MCM is based on 1377 precipitation gauges and on the mosaic of the Italian weather radars.

Hourly maps of the weather variables collected for the Po river basin ultimately include 1258 temperature stations, 608 for relative humidity, 460 for wind speed and 278 for solar radiation. Temperature maps include an altitude correction algorithm with temperature gradients estimated at every time step by linearly interpolating available data at different elevations. They

also include an outlier removal algorithm which discards station data with a deviation of more than 20°C from the corresponding temperature-elevation interpolating line.

Discharge data at 27 river gauging stations with hourly sampling frequency for the years 2016-2019 were provided by the DPC and the regional hydrometeorological offices. 22 stations were selected for model calibration, while 5 were retained for validation only (Figure 1). Validation stations were chosen to represent different areas of the Po basin, including a mix of small and large sub-catchments with varying influence of lakes and reservoirs.

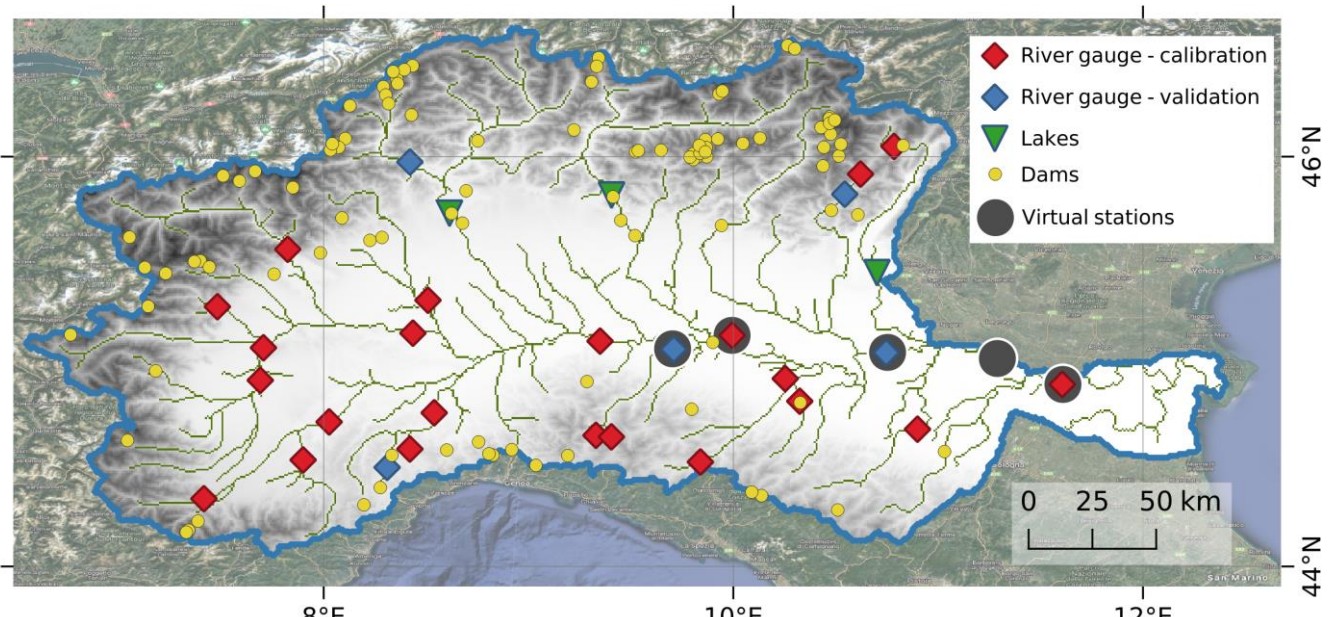

**Figure 1:** Simulated domain (blue line) and river network (dark green) of the Po river basin. Symbols show the point features considered
in the hydrological model.

## 2.3 Satellite products and validation

### 1.2.1 Precipitation

The precipitation dataset used in this work and referred to as SM2RAIN (Figure 2) merges SM2RAIN-ASCAT (Brocca et al., 2019) and the Global Precipitation Measurement (GPM) Mission IMERG-LR (Huffman et al., 2015) datasets, both
available at 10km spatial resolution. Unlike in Massari et al. (2020), where the fusion of the two datasets was based on an Optimal Interpolation Technique, here we relied upon a Triple Collocation (TC)-based merging using the Signal to Noise Ratio (SNR), as in Gruber et al (2017). In particular, to derive the merged dataset we seek the optimality in a least squares sense, so that the variance of residual random errors is minimized. This leads to a weighted average between SM2RAIN-ASCAT and IMERG-LR, i.e.,

$$P_{DTE} = w_1 P_{SM2RAIN-ASCAT} + w_2 P_{IMERG-LR} \qquad (1)$$

where the weights $w_1$ and $w_2$ are calculated as:

$$w_1 = \frac{SNR_1}{SNR_1 + SNR_2}, w_2 = \frac{SNR_2}{SNR_1 + SNR_2} \qquad (2)$$

SNR is estimated as the ratio between the variance of the true signal and that of the considered satellite product, multiplied by a parameter representing the systematic error (see Gruber et al. 2017), where the subscripts 1 and 2 refer to the SM2RAIN-ASCAT and IMERG-LR datasets, respectively. Under the assumption that the two datasets are independent (as also required by TC), the random error of the merged time series is lower than those of the individual input datasets.

TC was applied to the triplet: SM2RAIN-ASCAT, IMERG-LR and the MCM radar-gauge precipitation dataset. Note that, unlike the use of random error variances as in Crow et al. (2015), weights calculated as in (2) do not require the assumption of null systematic differences between the datasets, thanks to the self-consistency of the signal-to-noise ratio (see Gruber et al., 2017 for further details). Before the weights can be used to merge the data sets, relative systematic differences (i.e., long term bias) have to be corrected to make weights obtained by (2) converge to the optimal weights in a least square sense (Crow et al. 2015). Given the nature of the precipitation signal (containing many null values) this rescaling has been done by means of a multiplicative factor to the mean with respect to MCM. The fusion of the two datasets was only done for the time steps where IMERG-LR was greater than zero; due to the high sensitivity of the GPM mission, values with zero precipitation in IMERG-LR were set to zero. Hourly data were obtained by imposing the sub-daily temporal pattern of IMERG-LR to the merged dataset. The 10 km resolution dataset thus generated was resampled at 1 km resolution through bilinear interpolation for use in the hydrological model.

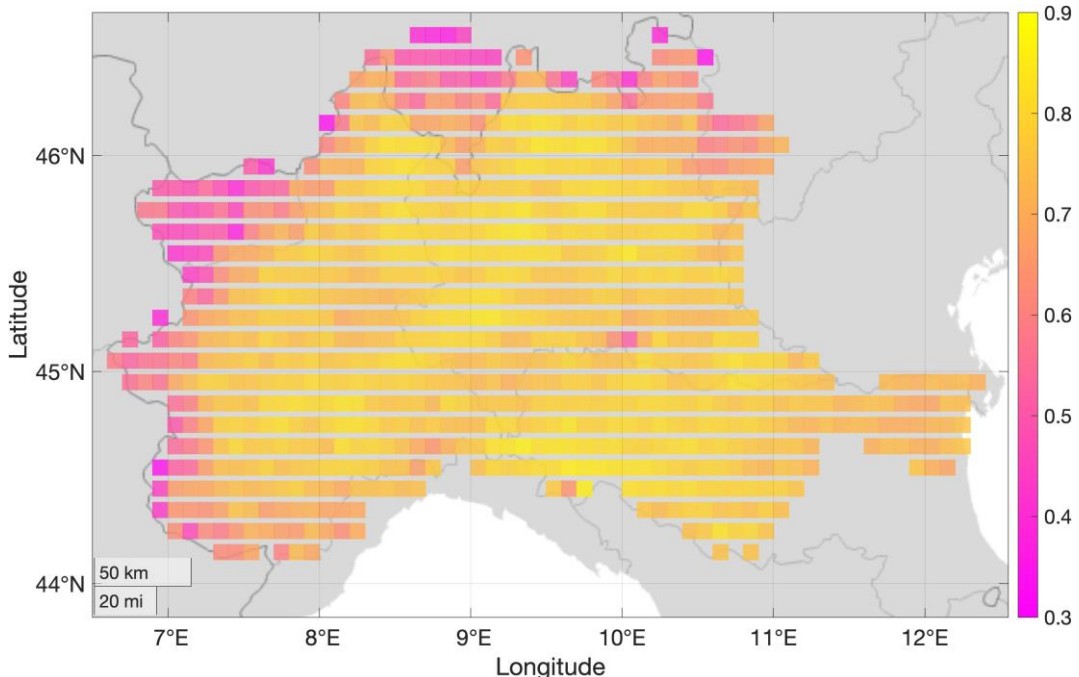

**Figure 2**: Daily Pearson correlation coefficient between SM2RAIN precipitation and the MCM (radar-gauge) precipitation dataset. Median correlation $r_{SM2RAIN}$=0.76 largely improves that of the two individual products, $r_{SM2RAIN-ASCAT}$=0.66 and $r_{IMERG-LR}$=0.67.

### 2.3.2 Evaporation

Global Land Evaporation Amsterdam Model (GLEAM, Miralles et al., 2011) is a state-of-the-art methodology to derive evaporation and its various components (i.e. transpiration, bare soil evaporation, interception loss, snow sublimation and open-water evaporation). It combines global satellite observations of meteorological (precipitation, near-surface net radiation, and air temperature) and surface (soil and vegetation water content, and snow water equivalent) variables that are informative for the evaporation process. The model is based on the Priestley and Taylor (1972) equation to estimate potential evaporation. Those estimates are then constrained based on root-zone soil moisture, which results from a precipitation-driven running water balance in which satellite-based soil moisture can be assimilated. Interception loss is independently estimated through an adapted Gash analytical model (Miralles et al., 2010). Since its first version, GLEAM has been widely deployed at coarse resolution for climatic studies. In the past few years, it has been further developed to solve higher spatial and temporal resolutions. For instance, Martens et al. (2018) obtained accurate results in an implementation over the Netherlands at 100-meter resolution. For this work, GLEAM was applied over the entire Po River Basin to produce both potential and actual evaporation estimates at 1 km resolution.

Since measurements of evaporation in the focus region are limited, the performance of the 1 km evaporation dataset was inferred on the basis of the FluxNet IT-Tor site, located in the mountainous Val d'Aosta region in the NW part of the domain (Figure 3). While based on one station only, the performance (Pearson's correlation $r$=0.83) is in line with results obtained in

the high resolution implementation across the Netherlands, where Martens et al. (2018) found a median temporal correlation coefficient of 0.76 across 29 sites.

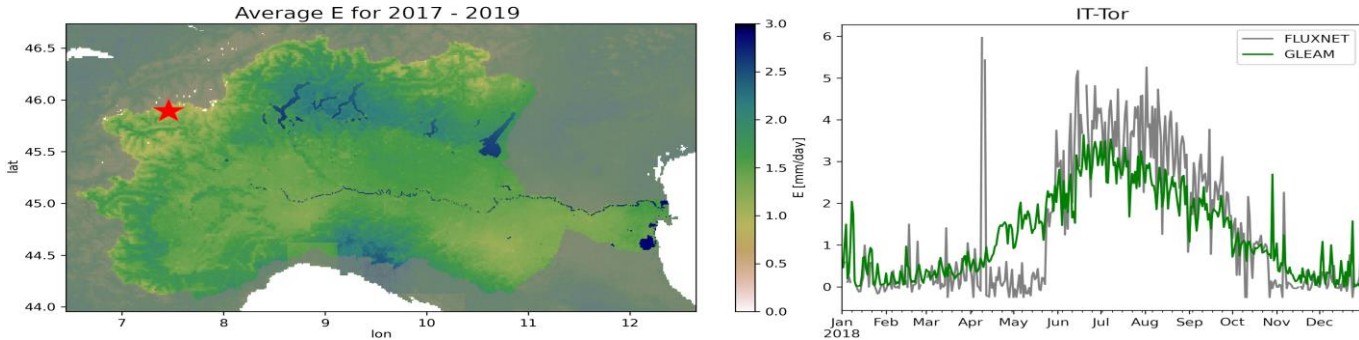


**Figure 3:** Average GLEAM daily actual evaporation for 2017-2019 in the Po river basin at 1 km resolution (left) and comparison with daily evaporation from the FluxNet site IT-Tor for 2018 (right). Pearson's correlation r=0.83. The location of the FluxNet site is marked with a red star on the left panel.

### 2.3.3 Soil moisture

High resolution soil moisture (SM) was retrieved from incidence angle dependent Sentinel-1 backscatter measurements at 500m spatial sampling (~1km spatial resolution) (Bauer-Marschallinger et al., 2019) by using a time series based first-order radiative transfer modelling approach (RT1, see Quast and Wagner, 2016; Quast et al., 2019). The RT1 model uses auxiliary Leaf Area Index (LAI) time series provided by ECMWF ERA5-Land reanalysis dataset (Muñoz-Sabater et al., 2021) to correct for effects induced by seasonal dynamics of vegetation. The retrieval is then performed via a non-linear least-squares
regression that optimizes static and dynamic model-parameters to minimize the difference between measured and modelled backscatter for a set of ~300,000 pixels over a 4-year time-period (2016-2019). The resulting soil moisture product represents a percentage measure of the relative moisture saturation of the soil surface. The performance of the obtained soil moisture time series was validated with in-situ observations as well as compared to top-layer (0-7cm) soil moisture estimates from ERA5-Land. In addition, the spatial distribution of the resulting auxiliary model parameters (single-scattering albedo,
soil scattering directionality) was analysed with respect to CCI Landcover (ESA, 2017) classifications to assess the physical plausibility of the resulting parametrization. The observed spatial pattern of the parameters indicate a close connection to the associated land cover, following some expected variations, e.g., higher single-scattering albedo over forested areas compared to croplands.

The RT1 high resolution soil moisture product over the Po basin shows an overall good performance compared to ERA5-
Land soil moisture, with a median Pearson correlation of 0.55 for croplands and 0.65 over areas primarily covered by natural vegetation (i.e., tree, shrub, herbaceous cover). Validation was performed using in situ soil moisture for the Oltrepo station (Bordoni et al., 2019) located in Canneto Pavese (PV, Italy) , which resulted in a correlation of 0.58 (raw data) and 0.73 (with a 10-daily rolling mean). These results highlight the potential of Sentinel-1 observations for high resolution soil moisture retrievals and their use in applied science.


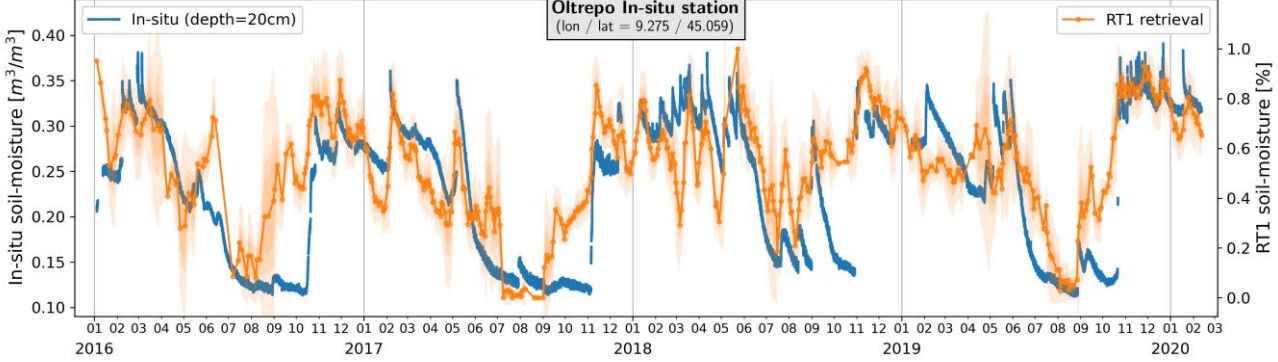

**Figure 4:** Time series of RT1 Surface SM compared to in situ SM in Oltrepo. A 10-daily rolling mean is applied to the RT1 retrievals to reduce noise. The shading indicates the corresponding standard-deviation. Pearson-correlation of 0.58 (raw data) and 0.73 (10-daily rolling mean).

### 215 2.3.4 Snow depth

Snow-depth data were obtained from the Sentinel-1-based product proposed by Lievens et al. (2019). The data product has a 1 km spatial resolution and daily granularity, and is available through the public repository of the C-SNOW project (https://ees.kuleuven.be/project/c-snow). The mapping algorithm is based on a change-detection approach and has been validated across the mountain regions in the whole Northern Hemisphere.


For the scope of the present study, C-SNOW data over the period September 2016 - April 2020 were evaluated with 172 ultrasonic snow-depth sensors across the Po river basin (Figure 5a). 77% of the evaluation dataset is located in the range 1000-2500 m above sea level (ASL) (Figure 5b), a frequent condition in the Alps (Avanzi et al., 2021a). Observed snow-depth data were processed by (1) setting to missing any negative value, (2) applying climatological thresholds for maximum

and minimum snow depth to remove spikes, and (3) using a threshold on the 6-hour moving coefficient of variation to detect periods with grass interference (Avanzi et al., 2014). Data was then aggregated at daily resolution, and C-SNOW data were extracted for the same locations and data range. The evaluation confirmed previous results by Lievens et al. (2019), with C-SNOW successfully reproducing the seasonality and magnitude of snow depth as measured by snow depth sensors (Figure 5c and d). Root mean square errors (*RMSE*) ranged from less than 20 cm below 1000 m ASL to 60 cm or more above 2000

m ASL, though with no significant trend in the bias versus the elevation.

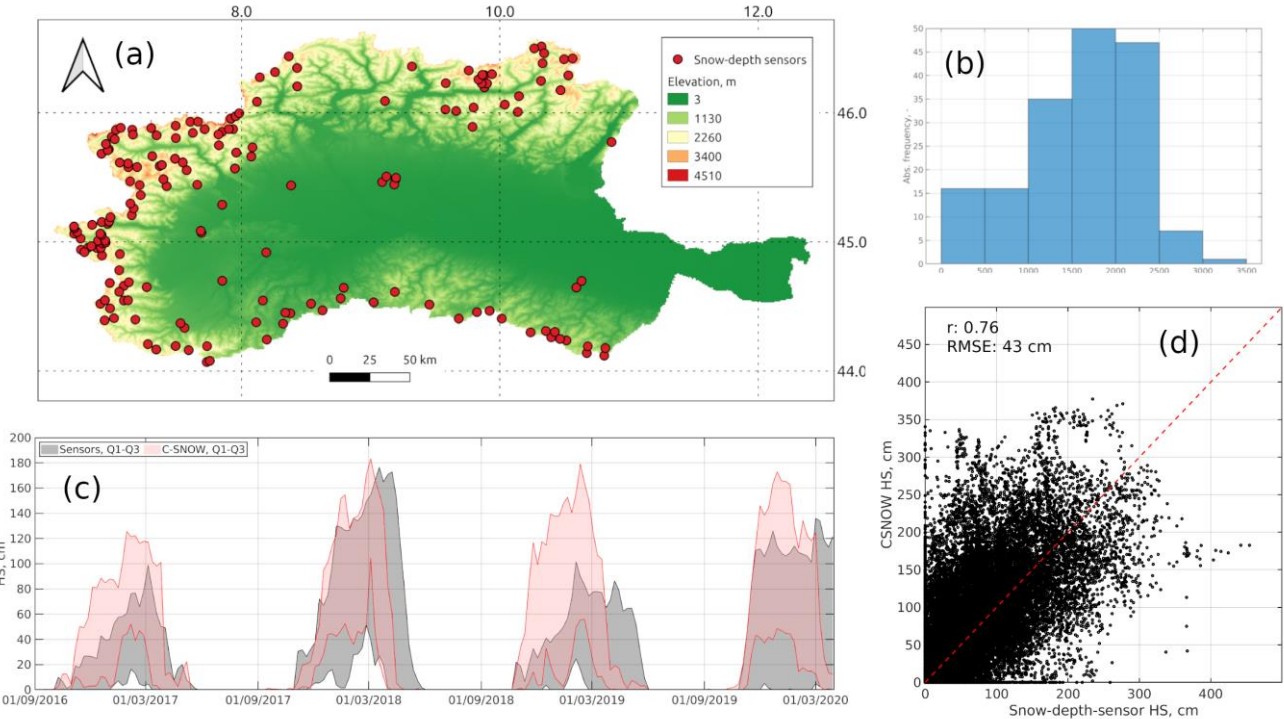

**Figure 5:** Evaluation of satellite based C-SNOW snow depth estimates. (a) Location of the 172 sensors across the Po river basin, and (b) their elevation distribution. (c) comparison between the interquartile range of C-SNOW and in situ measurements at the 172 sensors. (d) comparison between daily C-SNOW estimates and ground-based snow-depth measurements for all sites.

### 2.3.5 River discharge

River discharge time series from satellite remote sensing are estimated by integrating data from two sensors: altimeter and multispectral. Traditionally defined as the product of cross-sectional river flow area and velocity, river discharge is calculated by assuming that the satellite sensors measure the two quantities (Tarpanelli et al., 2015). Specifically, once the cross-sectional geometry is known, flow area is calculated as a function of the water height derived from satellite altimetry (Abdalla et al., 2021), while flow velocity, usually measured through in-situ instruments (current meter, acoustic doppler current profiler, velocimeter), is linked to the reflectance measured by the near infrared signal of the multispectral sensor (Tarpanelli et al., 2013), relying on the reflectance ratio between a dry (*C*) calibration pixel and the corresponding wet (*M*) measurement pixel.

Multi-mission satellite altimetry data coming from Saral/Altika, Cryosat-2 and Sentinel-3A and 3B are used to derive densified water level time series (Zakharova et al., 2020) at five stations along the main reach of the Po River named Piacenza, Cremona, Borgoforte, Sermide and Pontelagoscuro (i.e., virtual stations in Figure 1). At these stations, the multi-mission reflectance was extracted from the MODIS (Aqua and Terra), OLCI (Sentinel-3A) and MSI (Sentinel-2) sensors

following the methods shown in Tarpanelli et al. (2020). Here, river discharge ($Q$) is estimated as the product of flow velocity (Tarpanelli et al., 2020) and flow area, as a function of altimetry-derived water height ($H$) (Tarpanelli et al., 2015):

$$Q = \alpha(H)^\beta (C/M)^\gamma \tag{3}$$

where the parameters α,β,γ were calibrated using observed discharges at the five stations. The resulting time series for each station are illustrated in Figure 6 against the in situ observations recorded at the gauged stations. Performance metrics (Supplement material, Table S1) show skilful performance of the method in representing the observed daily discharges at the five stations, with average Nash-Sutcliffe (*NS*) of 0.81, *KGE* of 0.88 and relative *RMSE* (*rRMSE*) of 26%.

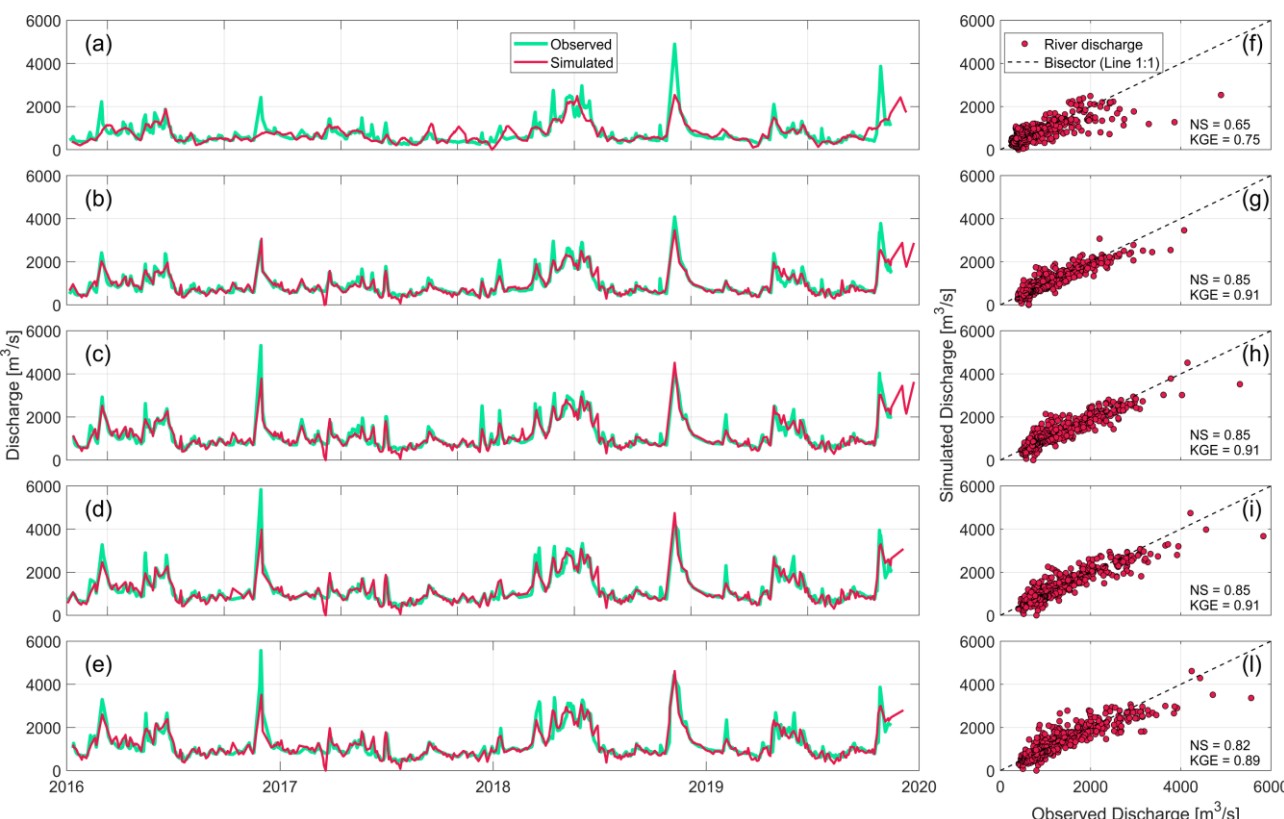

**Figure 6:** Comparison between discharges simulated by the multi-mission approach versus observations at five gauging stations in the Po River in terms of time series (left column) and scatter plot (right column):(a, f) Piacenza, (b, g) Cremona, (c, h) Borgoforte, (d, i) Sermide and (e, l) Pontelagoscuro.

## 3 Methods

### 3.1 Hydrological modeling

Continuum (Silvestro et al., 2013) is a distributed hydrological model relying on a morphological approach based on the identification of the drainage network components (Giannoni et al., 2000). It is a tradeoff between empirical and physically-based models, reproducing all main hydrological processes by relying on parameterization. The physical description of the

hydrological processes is comparatively simple, resulting in high computational efficiency yet generally skillful performance (Silvestro et al., 2013). Continuum reproduces the spatio-temporal evolution of runoff, soil moisture, energy fluxes, surface soil temperature, snow accumulation and melting. Evaporation is estimated through a bulk formulation by solving the  mass and energy balance as described in Silvestro et al. (2013) and related appendix, though it can also be provided as input variable. Deep flow and water table evolution are based on the Darcy's equation, where each cell drains towards the neighboring cells following the 2D water table gradient and their hydraulic head, while a distributed interaction between water table and soil surface is represented through parameterization. A force-restore equation (Dickinson, 1988) is used to model the surface energy balance and enables the estimation of land surface temperature.

To simulate the cryospheric processes, we used S3M version 5 (Avanzi et al., 2021b), a one-layer snow model accounting for precipitation-phase partitioning, snowpack accumulation and melt, snow rheology and hydraulics, as well as glacier melt (Terzago et al., 2020; Avanzi et al., 2021b). With its hybrid approach to snowmelt, which decouples the radiation- and temperature-driven contributions, S3M combines a parsimonious formulation with a substantial physical realism. For this work, S3M and Continuum were set up and run over the entire Po River basin (drainage area of 74,000 km$^2$), with a constant grid spacing of 1 km and time resolution of 1 hour.

### 3.2 Model calibration

To improve the representation of the hydrological states, Continuum was calibrated in the focus region using discharge data as benchmark. We deployed a multi-site calibration procedure that iteratively searches the model parameterization that best matches the available discharge observations over the calibration period at the 22 considered calibration stations (Figure 1), through minimization of a cost function. Hydrological simulations run for the model calibration cover the 2 years starting on 2018-01-01, while the calibration period starts on 2018-07-01, leaving out the initial 6 months for model warm-up. The calibration tool perturbs six scalar parameters related to four physical hydrological features: infiltration velocity at saturation (*cf*), field capacity (*ct*), Curve Number (*CN*), and water sources (*ws*).

While the calibrated value of *ws* is a constant for the entire region of interest, for *ct*, *cf* and *CN*, the calibration consists in a rescaling of their default maps to the best value, thus preserving their spatial pattern, which depends on geographic spatial datasets of soil characteristics and land cover. The cost function, based on the Kling-Gupta Efficiency (Gupta et al., 2009), computes an error between the duration curves at each percentile, weighted with the logarithm of the upstream area, to give higher weight to the downstream stations without neglecting the contribution of the most upstream ones.

The calibration procedure was performed through the implementation of a parallel search algorithm. The algorithm performs an iterative exploration of the 6-dimensional parameter space; the exploration starts with *N*=20 initial values sampled with a Gaussian Latin Hypercube approach. For each of these *N* parameter sets, a hydrological simulation is performed over the calibration period, and the cost function is computed to map the error hypersurface. The point that minimizes the cost function is used as the centre of the following iteration, until the algorithm converges to an optimal solution.

### 3.3 Data assimilation of satellite snow and soil moisture products

Satellite derived soil moisture from the Sentinel-1 RT1 product was assimilated into the Continuum model through a nudging technique (Stauffer and Seaman, 1990; Lakshmivarahan and Lewis, 2013). The nudging scheme is a computationally inexpensive approach and is particularly suitable for applications in operational frameworks for flood predictions. The update is performed when the satellite data become available, on average once per day for soil moisture, following the equation:

$$X_{MOD}^{+}(t) = X_{MOD}^{-}(t) + G[X_{OBS}(t) - X_{MOD}^{-}(t)] \tag{4}$$

where $X^{+}_{MOD}$ represents the updated modelled variable , $X^{-}_{MOD}$ is the modeled prior value, $X_{OBS}$ is the observation, and $G$ is the Kernel function. Thus, the correction term represents the difference between observed ($X_{OBS}$) and modelled variable multiplied by $G$ that takes into account the uncertainties of both model and satellite observations. In this application we used a constant value of $G$=0.45, following the recommendations by Laiolo et al. (2016), who estimated optimal $G$ values from a test on four different satellite-derived soil moisture products. In addition, we used $G$=0 in areas with low Pearson correlation coefficient ($r$<0.7) between satellite-derived and modeled soil moisture in the simulation period.

The assimilation of satellite-derived C-SNOW maps into S3M was performed using the same approach and assuming $G$=1 to mimic direct insertion. C-SNOW maps come as snow depths, while S3M supports assimilation in the form of snow water equivalent (SWE), which is a more suitable variable to assimilate to control the water balance. Thus, snow depths from C-SNOW were converted in SWE using simulated snow density values (see Avanzi et al., 2021b). Along with snow depth information, we rely on C-SNOW to determine snow-covered and snow-free areas, and then assimilated this information into S3M to clip modeled snow cover according to the satellite information. More information on the theoretical background of SWE assimilation in S3M can be found in Avanzi et al. (2021b).

## 4 Results

### 4.1 Baseline run

The hydrological model Continuum was first calibrated using conventional meteorological data and observed discharges at the 22 calibration stations described above. The calibrated setup was then run over the years 2016-2019 to produce a baseline simulation for 2017-2019, leaving out 2016 as model warm-up. Average evaporation in 2017-2019 computed by Continuum for the Po basin is 950 mm/year and results 21% smaller than the GLEAM average of 1200 mm/year in the same time span. A comparison of simulated versus observed hourly river discharges is shown in Figure 7 for five sample stations, while six performance metrics are shown for all 27 discharge stations in Figure 8 and in Table S2 (see Supplement material). Dimensionless scores, including *KGE* and its three decomposition terms, i.e., correlation (*r*), bias rate, and Coefficient of variation rate (*CV* rate), increase on average with the upstream area. Note that all four scores have optimum value at 1. The mean *KGE* over all the stations *KGE*=0.51, rises to 0.63 and 0.70 for basins larger than 1,000 km$^2$ and 10,000 km$^2$,

respectively. Similar trends versus the same classes of upstream area are found in the mean correlation (0.75, 0.86, 0.88), while bias rate (0.98, 0.99, 0.94) and *CV* rate (0.89, 0.94, 1.13) are slightly deteriorated for basins larger than 10,000 km$^2$. Differences in the mean *KGE*, *r* and *CV* rate between validation and calibration stations are not statistically significant in a two-sample t-test for the mean. Only the mean of the bias rate of the two samples is statistically different at 5% significance level, with validation stations having an average 30% negative bias in comparison to an average 5% positive bias of the calibration stations.

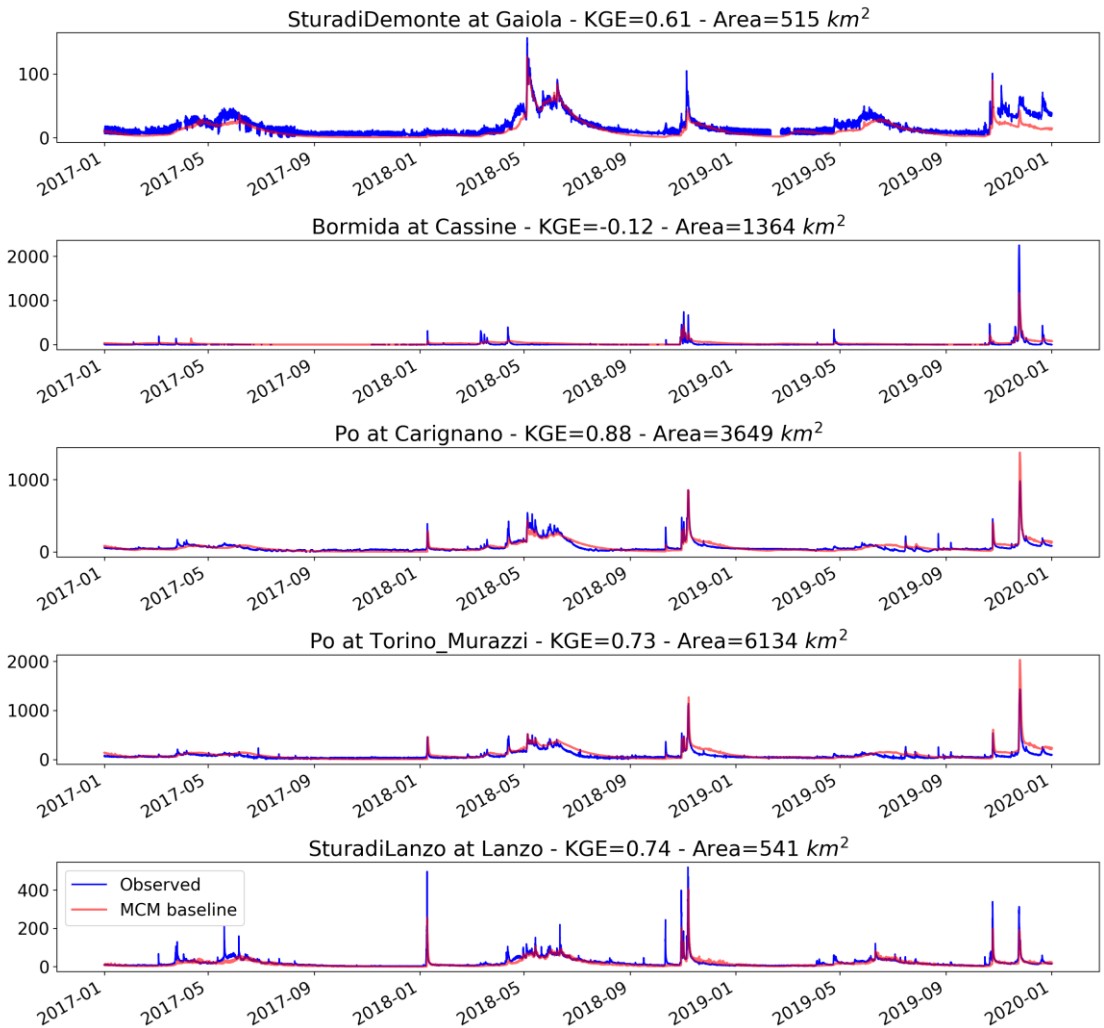

**Figure 7:** Observed versus simulated (baseline) discharge [m$^3$/s] for the years 2017-2019 at five river gauging stations.

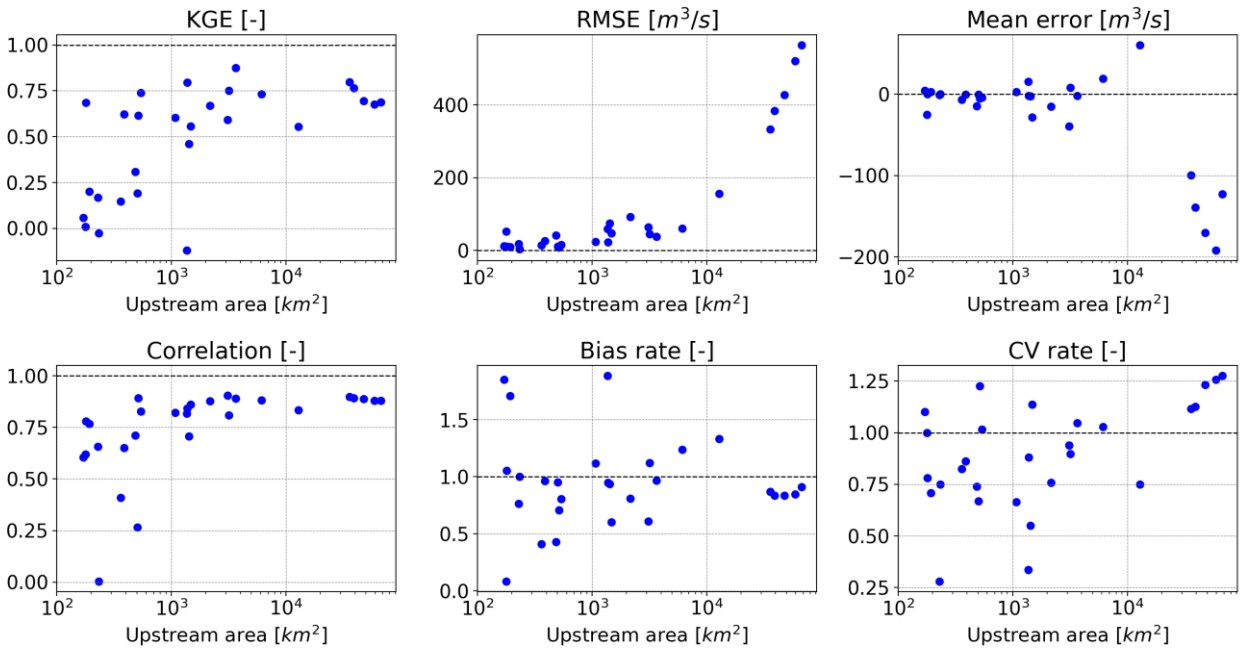

**Figure 8:** Skills of the baseline run versus upstream area at the 27 measurement stations. Dashed lines indicate the optimum value of each score.

## 4.2 Model runs with satellite input

In a second phase, we performed four hydrological simulations, each of them based on the configuration and input data of the baseline run and by replacing in turn one input dataset with one of four satellite products described in Sect. 2.3: 1) precipitation from SM2RAIN, 2) evaporation from GLEAM, data assimilation of 3) soil moisture from RT1, and of 4) snow depths from C-SNOW. Two additional configurations were run including multiple satellite-based data sources: 5) all four satellite Earth observation datasets, hereafter referred to as EO, and 6) a combination of the satellite precipitation and evaporation, referred to as SM2RAIN+GLEAM. The spatial distribution of the performance of the six model simulations at the 27 river gauges is shown as maps of *KGE* (Figure 9) and its three decomposition terms (see Supplement material). Further, boxplots of the *KGE* of the six experiments and comparison with the baseline run are shown in Figure 10.

Results denote a generally skillful reconstruction of river discharges for all experiments, with mean *KGE* at the 27 stations ranging between 0.13 (SM2RAIN+GLEAM) and 0.53 (C-SNOW), all well above the no-skill threshold of $KGE_0 = 1\text{-}2^{1/2} \cong \text{-}0.41$ (see Knoben et al., 2019). Simulations including C-SNOW and GLEAM perform on average better than the baseline run, with mean improvements in *KGE* of 0.02 and 0.01 (+4% and +2%), respectively. Largest differences in the overall performances are due to the wide range of the mean bias across the six simulations, with the largest bias rates for SM2RAIN+GLEAM (1.58) and EO (0.69), and the lowest bias rate for GLEAM (1.02) and C-SNOW (0.97), both improving that of the baseline run (0.95). On the other hand, average correlations across the six experiments fall in a much narrower

interval, ranging between 0.61 for EO and 0.75 for both C-SNOW and the baseline run. Running the model with all EO data produces on average a 28% deterioration of the mean performance (*KGE*=0.37), though it surprisingly generates the best performance at the five validation stations (*KGE*=0.54) among all simulations (Figure 10).

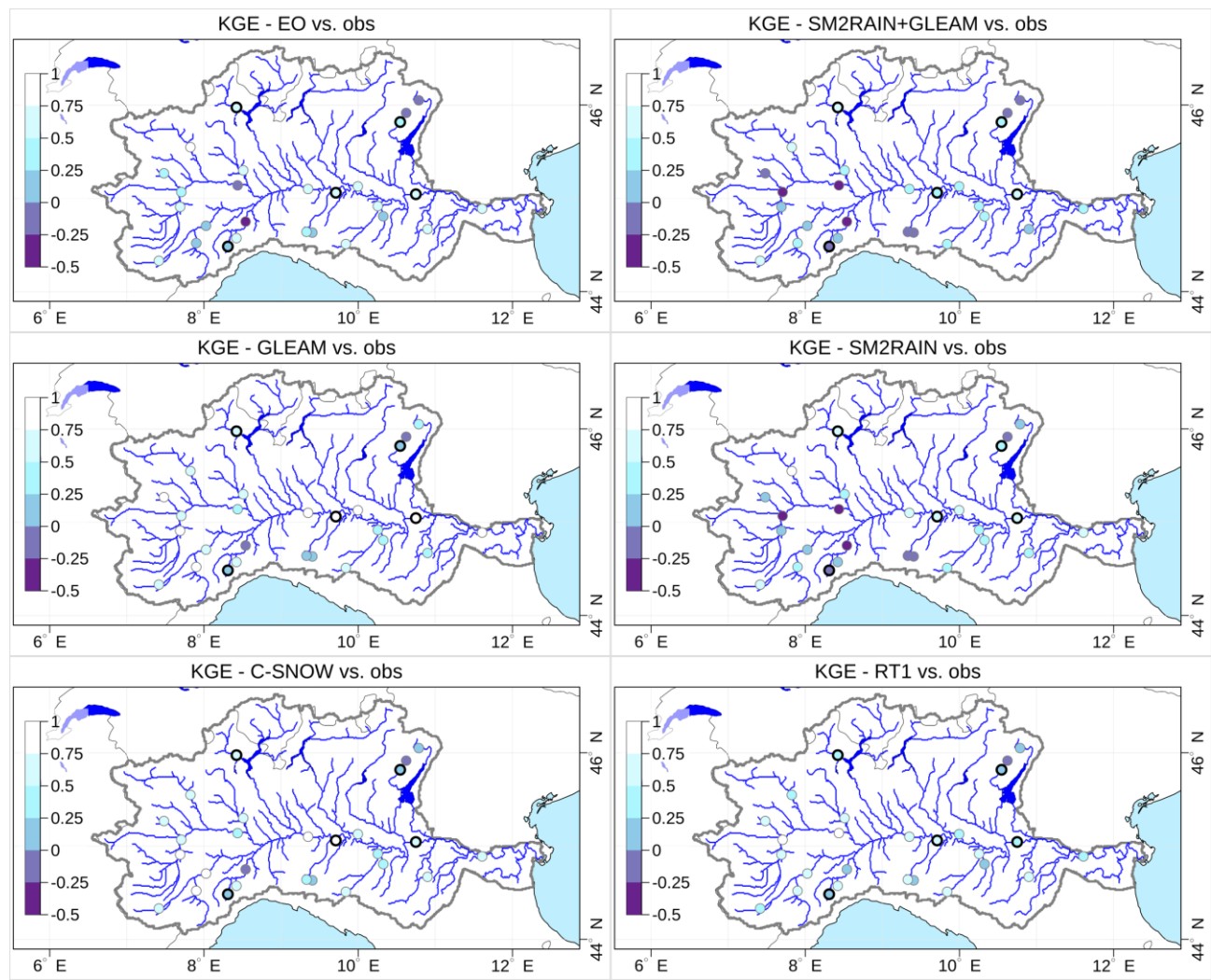

**Figure 9:** Spatial distribution of the Kling-Gupta Efficiency (*KGE*) of the six model runs driven by the four input satellite products versus observed discharges at the measurement stations. Validation stations are marked with a bold circle. Multi-product experiments are in the first row, while single-product experiments in row 2 (forcing input) and 3 (data assimilation input).

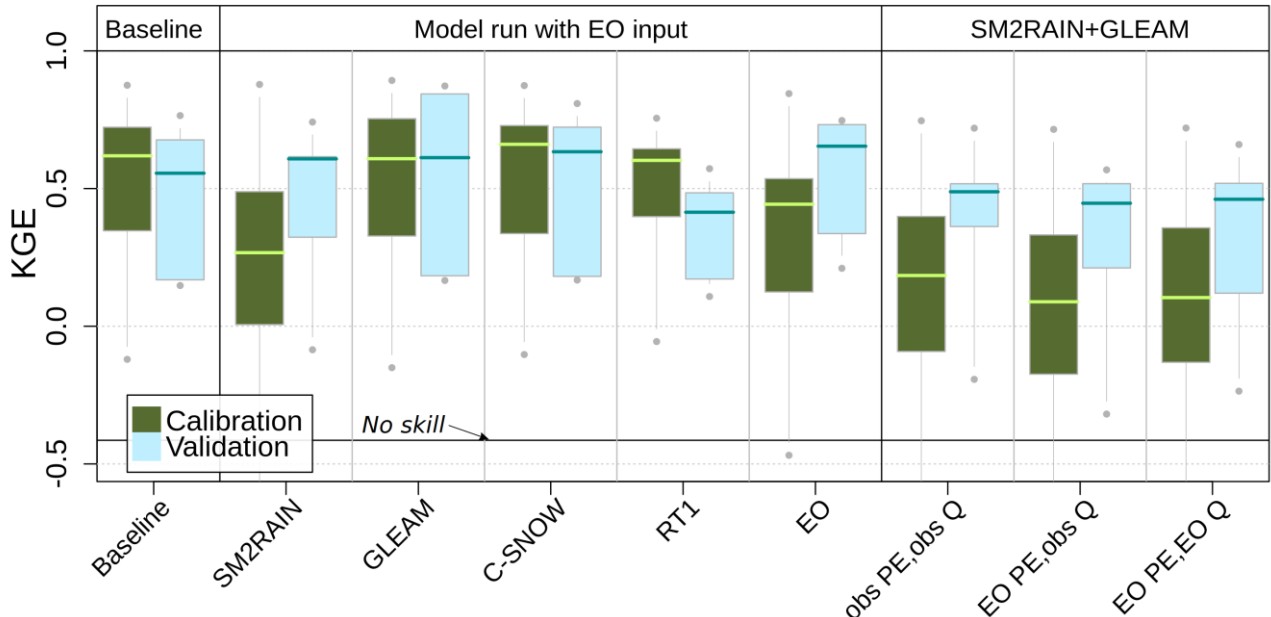

**Figure 10:** Box-plots comparing the Kling-Gupta Efficiency (*KGE*) of simulated river discharges for all the considered experiments versus observations at the calibration and at the validation stations. The no-skill line at $1-2^{1/2}$ is indicated with a solid horizontal line. In the three rightmost columns, PE stands for precipitation and evaporation, while Q stands for discharge.

The six simulations forced by satellite products were then compared to the baseline run, to detect similarities and deviations in the entire simulation domain, including where no observations are available. To reduce the correlation effects along the river network, we consider only one value per simulated river reach, located just upstream each confluence. Using RT1 and GLEAM does not result in significant spatial differences with respect to the baseline (Figure 11). As expected, the use of C-SNOW results in differences mainly in alpine areas, especially in Ticino (Switzerland), where the MCM dataset used in the baseline run is known to underestimate precipitation rates, due to the lack of ground measurements outside the Italian territory. Larger deviations are visible in the runs including SM2RAIN, particularly in the upper Po basin in the west and in the upper Adda River in the north, confirming the stronger sensitivity of river discharge to precipitation dynamics.

Figure 12 shows a comparison of the six simulations forced by satellite products, the baseline run, and observed discharges at two validation stations, for a series of moderate to high intensity events which hit a large portion of the Po River basin in Fall 2019. The second of the three main events, in the second half of November, caused the exceedance of the maximum alert level and widespread flooding in several river sections in the main reach of the Po River across the Lombardia and Emilia Romagna regions, including the area of Piacenza (Figure 12, bottom). In Piacenza, all model simulations performed reasonably well, with maximum error on the peak discharge below 20%. The best performances over the three months are found in the baseline run and in the two runs with data assimilation (RT1 and C-SNOW), all three with *KGE*=0.89. Lower performances are produced by the three runs forced by SM2RAIN, mainly due to an overestimation of the first event in late October 2019. At the Candoglia station, results show an opposite pattern, with best performance by SM2RAIN and

SM2RAIN+GLEAM both with *KGE*=0.74 over the three months, mildly improving upon the performance of the baseline run (*KGE*=0.71).

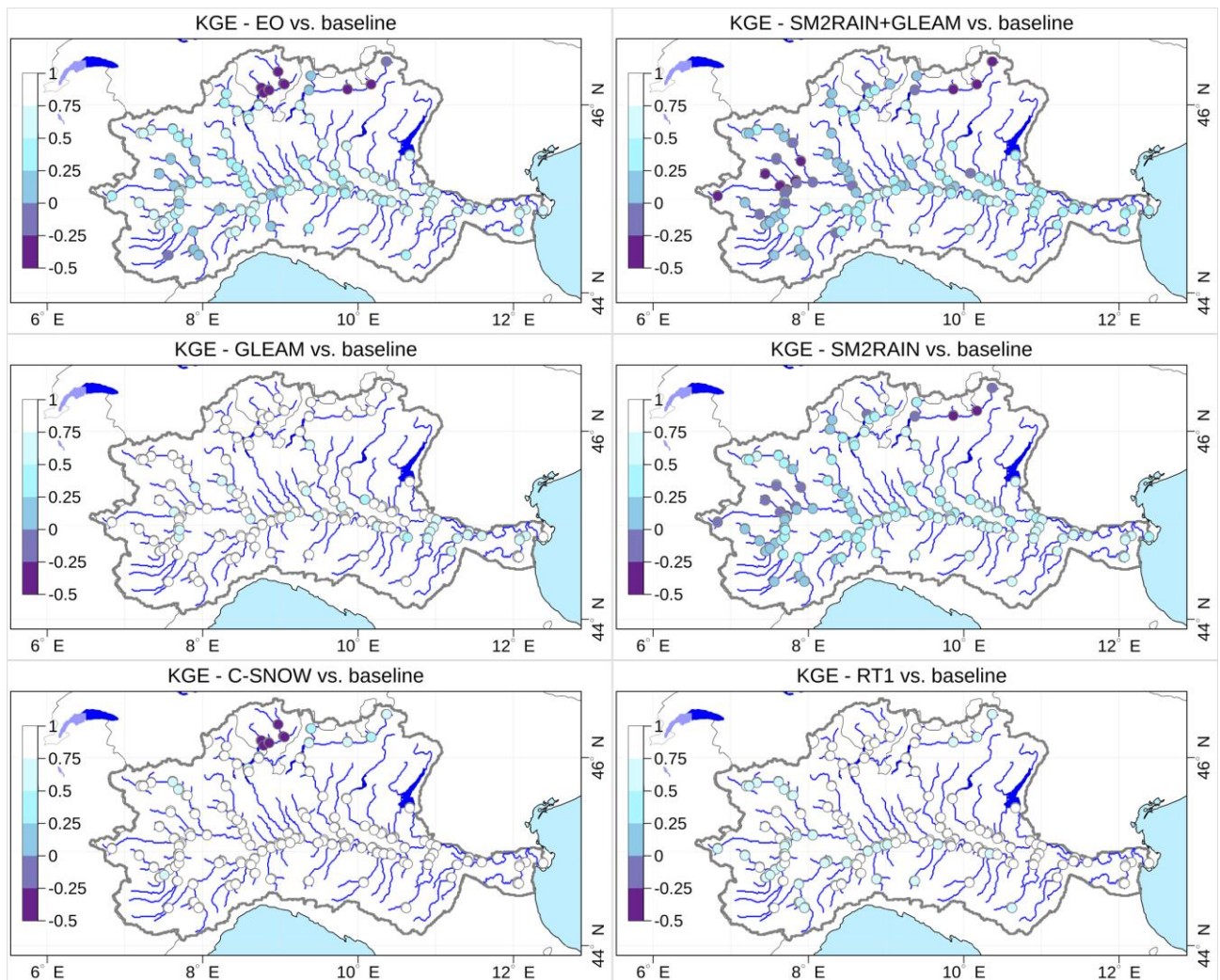

**Figure 11:** Spatial distribution of the Kling-Gupta Efficiency (KGE) of discharges of the six model runs driven by the four input satellite products versus the baseline run, at each modeled river reach. Multi-product experiments are in the first row, while single-product
experiments in row 2 (forcing input) and 3 (data assimilation input).

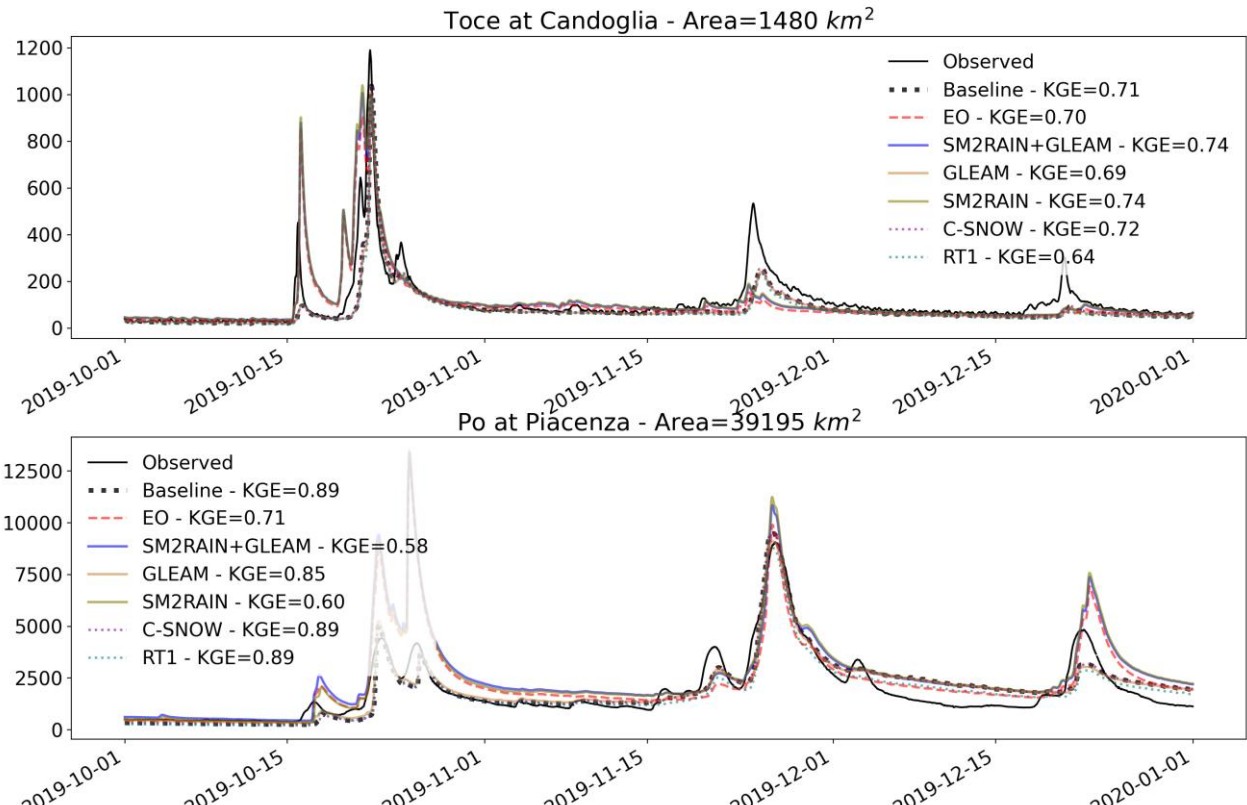

**Figure 12:** Comparison of observed and simulated hydrographs [m³/s] for the events of October-December 2019 at two validation stations: Candoglia (top) and Piacenza (bottom), together with *KGE* calculated versus the observed discharges for the same three months.

## 4.3 Sensitivity of satellite data to three model parameterizations

A subsequent experiment investigated the performance of the hydrological model in reproducing discharges at the 27 river gauges, by forcing it with the satellite datasets SM2RAIN and GLEAM. In details, we compared the results of three model runs over 2017-2019, using three different model parameterizations obtained through dedicated calibrations (over 2018-2019), derived by applying the steps described in Sect. 3.2 to different configurations of input and benchmark discharges:

1. The first is the simulation SM2RAIN+GLEAM described in Sect. 4.2, i.e., run with the model parameters obtained by calibrating with conventional ground observations (interpolated measurements and MCM precipitation) and optimizing the objective function using observed discharge at the 22 calibration stations as benchmark (obs PE, obs Q in Figure 10).

2. The simulation SM2RAIN+GLEAM run on a model calibration forced by the same satellite datasets SM2RAIN and GLEAM as input and optimizing the objective function using observed discharge at the 22 calibration stations as benchmark (EO PE, obs Q in Figure 10).

3. The simulation SM2RAIN+GLEAM run on a model calibration forced by the satellite datasets SM2RAIN and GLEAM as input and optimizing the objective function using satellite-derived discharge estimates at the 5 virtual stations (see Sect. 2.3.5) as benchmark (EO PE, EO Q in Figure 10).

It is worth noting that soil moisture and snow depth data were not used in this experiment because they are not model input variables but rather assimilation variables, hence the calibration procedure described in section 3.2 would not be directly applicable. Results from simulation #2 forced by the same SM2RAIN and GLEAM used in the calibration shows the lowest performance among the three (mean *KGE*=0.07 over all 27 stations). Simulation #3 (EO PE, EO Q) gives satisfactory performance (mean *KGE*=0.10), relatively close to #1 (mean *KGE*=0.13), despite relying largely on satellite data.

Interestingly, the five validation stations on average outperform the set of calibration stations, with average *KGE* of 0.38, 0.30 and 0.29 for the three experiments. Performance of the three model runs versus the upstream area at the 27 stations (Figure 13) shows a general improvement in the correlation with the upstream area, while for the other metrics trends are less clear. Simulation #3 shows reduced variability (*CV* rate) yet smaller absolute errors (*RMSE* and mean error in Figure 13), also thanks to a calibration focused on the downstream virtual stations.

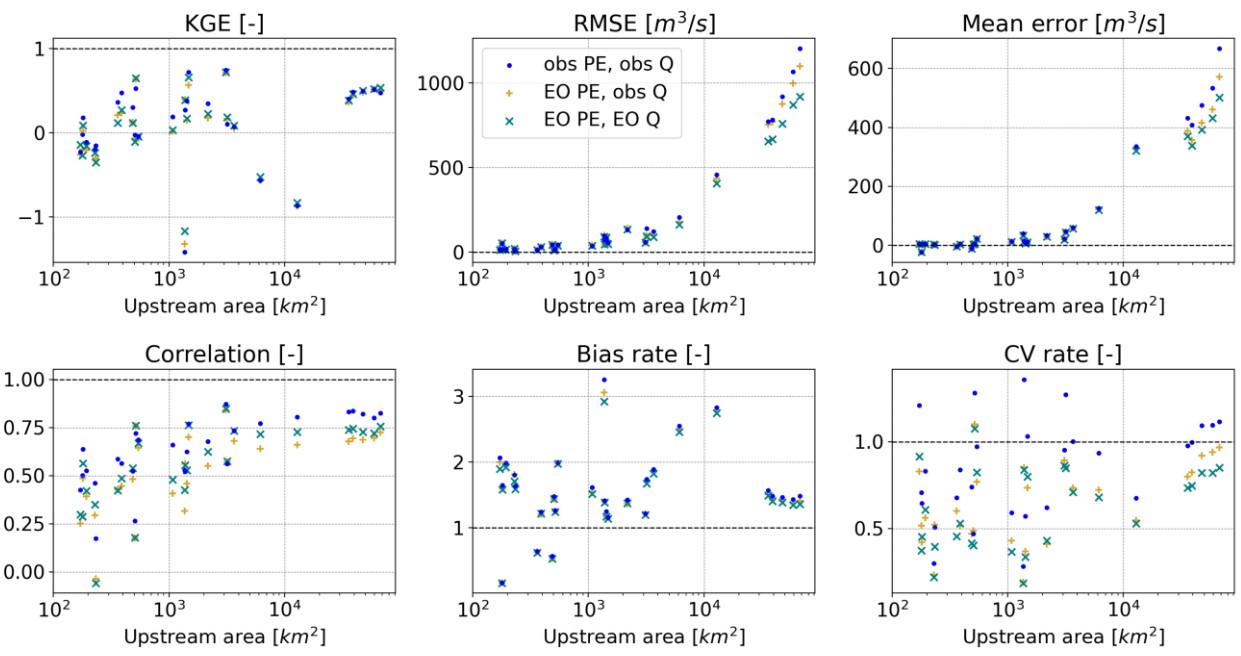

**Figure 13:** Skills of the run forced by satellite precipitation and evaporation (PE) versus upstream area at the 27 measurement stations. The three markers denote three calibrated parameter sets, obtained with different configurations of PE input and of benchmark discharge (Q). Conventional observational datasets are indicated with "obs", while "EO" are the satellite-derived datasets. Dashed lines indicate the optimum value of each score.

## 5 Discussions

A critical evaluation of the results of the experiments performed can help identify strengths and weaknesses, as well as directions to take to maximize the benefits of satellite observations in Earth system modeling. Overall, hydrological simulations driven by satellite datasets produced encouraging results, with 95% of the *KGE* of the station-experiment combinations above the no-skill threshold (versus 100% for the baseline run). The remaining 5% of combinations with *KGE* below the no-skill threshold occur in just 3 stations out of 27 and only in model configurations including SM2RAIN. Generally, the precipitation dataset is found to have the largest weight on the resulting model performance, with standard deviation of changes in *KGE* versus the baseline simulation $SD_{\Delta KGE,SM2RAIN}=0.37$ being more than twice that of all the other satellite driven configurations ($SD_{\Delta KGE,RT1}=0.16$, $SD_{\Delta KGE,GLEAM}=0.09$, $SD_{\Delta KGE,C-SNOW}=0.06$). In other words, the simulation performance shows strongest sensitivity to the precipitation forcing, which in fact leads to the largest deteriorations compared to the baseline run, as well as some of the largest improvements in *KGE*, up to $\Delta KGE_{MAX,SM2RAIN}=0.29$, well above all the improvements produced by GLEAM ($\Delta KGE_{MAX,GLEAM}=0.17$) and C-SNOW ($\Delta KGE_{MAX,C-SNOW}=0.12$) at any single station. This result is largely in agreement with previous findings (e.g., Jones et al., 2006; Sperna Weiland et al., 2015) and highlights the importance of advances in satellite precipitation estimation for hydrological applications. Qi et al. (2016) showed that model performance can also be impacted by model-precipitation product interactions, though this can partly be mitigated by dedicated model calibrations for each combination of input products. The high resolution version of SM2RAIN used in this work leads to comparable hydrological performance to that of the best non-gauge-corrected satellite products found in the literature (Camici et al., 2018; Amorim et al., 2020), and local results are better than those obtained with previous coarser resolution versions (see e.g., Beck et al., 2017; Tang et al., 2020). These works also show that satellite precipitation datasets bias-corrected with ground observations further improve the overall quality, including the performance in hydrological modelling.

With regard to the precipitation forcing, one must also note that the MCM dataset used in the baseline represents a particularly difficult benchmark to overcome. The high station density and the merging with the Italian radar composite make MCM a high-quality and detailed product both spatially and temporally. Yet, only few world areas can rely on seamless and nearly unbiased gauge-radar products, while satellite datasets remain prime candidates in ungauged regions, especially for real time applications, thanks to key features such as extended coverage, high resolution, short latency, and spatial consistency. In addition, satellite datasets are unaffected by country borders, which make them suitable for applications in transboundary river basins, especially in countries where data sharing agreements are not easily implemented. In contrast, GLEAM and C-SNOW consistently produced moderate improvements, though on a larger number of river sections, with only a minority of stations where skills deteriorated in comparison to the baseline run. Finally, the assimilation of RT1 soil moisture shows contrasting behaviour. On the one hand, it deteriorated *KGE* values throughout most of the stations in the main reach of the Po river, due to a general increasing negative bias. On the other hand, it shows general benefits in small-size upstream catchments and notably the best improvement in *KGE* ($\Delta KGE_{MAX,RT1}=0.41$) among all 216

station-experiment combinations, for the Trebbia river at Valsigiara. Our findings confirms the challenges in implementing a semi-automated assimilation of satellite soil moisture already pointed out in previous research (Laiolo et al., 2016; Wanders et al., 2014), where a range of factors affect and often decrease the assimilation performance, including the presence of complex topography, snow cover, frozen soil, urban areas, as well as differences between modeled and actual vegetation cover and leaf area index.

A final comment goes to the surprisingly high skills of hydrological simulations at the 5 validation stations, which on average exceed those at the calibration stations in 5 out of 9 experiments (see Figure 10). The multi-site calibration strategy is designed to find an optimal parameter set for the entire domain, thus reducing the effect of highly variable model performance typical of cascading calibrations (e.g., Alfieri et al., 2020). All results are then compared at the calibration and validation stations for the same period 2017-2019, which is twice the duration of the calibration period, implicitly adding a validation component also at the calibration stations. Higher performance at the validation stations seem to be particularly evident in simulations forced by SM2RAIN, though a connection between these facts is not known and it may simply be related to spatial differences in the skills of the satellite precipitation forcing in the sub-catchments where validation stations are located. A noteworthy case is that of the validation station of the Toce River at Candoglia, in the north-western part of the Po basin. It is influenced by a large number of reservoirs upstream and the Lake Maggiore located just downstream hugely smoothens its runoff characteristics from the rest of the river network. This makes the sub basin almost disconnected by the rest of the Po basin. Notwithstanding, simulation performance at Candoglia are higher than those of the calibration stations in all experiments but one (RT1), with the case of SM2RAIN scoring a *KGE*=0.74, hence 0.22 points higher than the average calibration *KGE* among all stations. Given the number of experiments presented, focused on the role of different input data and model parameterization, results are only shown through overall statistics of each model run. Future work will investigate detailed model behavior over specific hydrological processes, regimes, seasonality and quantiles of the flow duration curve, to better disentangle strengths and weaknesses of the considered satellite products in specific hydrological conditions.

**6 Conclusions**

This research explored the impact of five high resolution satellite products in distributed hydrological modelling. In a set of experiments we tested the use of satellite precipitation and evaporation as forcing input, data assimilation of satellite soil moisture and snow depth, and satellite river discharge estimates as benchmark for model calibration. We found skilful performance for all simulations including satellite derived products, with GLEAM evaporation and C-SNOW snow depth yielding an average 2% and 4% improvements over a baseline run driven by high-quality ground-based datasets. The skills of model runs including Earth observation data showed considerable variability in space and time. In addition, we found skilful results in a model calibration heavily relying on satellite products, both with regard to forcing input and to benchmark

discharge. This heralds the use of hydrological models fully relying on satellite data as an appealing solution for large scale applications and for regions where ground-based observations are not available, particularly in near-real time.

*Code and data availability.* Satellite products developed for this work can be requested from the authors. S3M and Continuum are open source models and their code is available at https://github.com/c-hydro.

*Author contributions.* LA, LB, SG conceptualized the work; FA, FD, CM, LB, AT, DR, DGM, RQ, MV prepared and validated the satellite products; FA, FD, LC, AL developed model code for the analysis; LA, GB and AL prepared the model input data; LA performed the formal analysis, data visualization and wrote the draft article with the contribution of all the authors.

*Competing interests.* The authors declare no conflict of interest.

*Acknowledgements.* The research leading to these results has received funding from the European Space Agency (ESA) contract n. 4000129870/20/I-NB (CCN. N.1) "Digital Twin Earth (DTE) - Hydrology". Observed snow depths were provided by the Italian Civil Protection Agency and the administrative regions of Valle d'Aosta, Piemonte, Lombardia, Liguria, Veneto, and Emilia Romagna. DGM acknowledges support from the European Research Council (ERC, grant no. 715254 DRY–2–DRY).

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
