# Peer review of "High resolution satellite products improve hydrological modeling in northern Italy"

_Hydrology and Earth System Sciences, 2021_

## Referee Comment (RC2)

Comments on 'High resolution satellite products improve hydrological modeling in northern Italy' by Alfieri et al.

This manuscript reports recent advances in the use of high-resolution satellite-based Earth observation data in hydrological modelling. A distributed hydrological model called the Continuum is used for the Po River Basin in a set of experiments using satellite precipitation and evaporation as forcing and assimilating satellite-derived soil moisture and snow depths. The general description of the experiments and the used datasets as well as evaluation of the simulation results are adequate. However it is felt that more technical details should be presented in order to fully understand the presented results.

1. While the results are presented in terms of statistical measures, the goodness in simulating high flows and low flows are obviously very different for different gauging stations. This is a major issue that needs attention and at least a discussion and some additional supplementary materials are required.

2. L88-89, the ECOCLIMAP (2013) was used for the vegetation coverage. There are obviously more recent data for the period of the simulations. A comparison and quantification of the uncertainty is needed.

3. The MS needs to report the used signal to noise ratio that determine the weights in eq. (1-2).

4. Datasets of different spatial resolution are mentioned, but what is the used spatial resolution in the hydrological modeling, 10km? How are they converted to the same resolution?

5. L163-165: The used RT1 model for soil moisture retrieval uses auxiliary Leaf Area Index (LAI) time series provided by ECMWF ERA5-Land reanalysis dataset to correct vegetation effects, but ERA5-Land assumes a fixed land cover and static monthly leaf area index (LAI) climatology. How does the actual LAI change, in particular in agricultural areas, impacts the soil moisture and the subsequent use of it in the hydrological modeling?

6. In eq. (4), G is assumed as 0.45, meaning the observation carries 45% information and the simulated background 55% information for all grids. This seems a gross simplification for different land cover types and needs at least a discussion and justification.

7. L314-316: '*Results denote a generally skillful reconstruction of river discharges for all experiments, with mean KGE at the 27 stations ranging between 0.13 (SM2RAIN+GLEAM) and 0.53 (C-SNOW), all well above the no-skill threshold of KGE0 = 1-21/2 $\cong$0.41 (see Knoben et al., 2019).*' Please explain how 0.13 > 0.41?

---

## Author Response (AR1)

MS No.: hess-2021-632

Title: High resolution satellite products improve hydrological modeling in northern Italy

Dear editor and editorial support team of HESS,

We would like to thank you and the reviewers for the positive evaluation of the article we presented. In the revised version we have further improved the article following the reviewers' recommendations and their helpful comments. We do not disagree with any of the reviewer's comments so, the vast majority of those have resulted in an addition to the text or to a change. Three figures have been remade according to the reviewers' suggestions and are attached to this submission. Our reply to each comment is shown below, interspersed with the reviewer's comments.

Reply to Referee #1

We thank the reviewer for his/her time in reading our manuscript and for the overall positive evaluation received. We do not disagree with any of the reviewer's comments so the vast majority of those have resulted in an addition to the text or to a change. Our reply to each comment is shown below, interspersed with the reviewer's comments.

General comments

I have troubles with the GLEAM data. Please state very explicit if you are using PET (potential ET) or AET (actual ET) and for what purpose. I assume Fig3 shows AET, but as model forcing you likely use PET. The spatial pattern in Fig3 is hard to interpret as I am not sure about what variable is shown. GLEAM estimates AET and I wonder why it has not been included in the data assimilation (or calibration), similar to SM and snow depth? Please be more explicit and state that the GLEAM scenario uses PET from GLEAM. If I understood this correctly.

Reply:

We agree with the reviewer and added some details to improve the understanding on the use of evaporation data. We have added to the manuscript that "Alternatively, both actual and potential evaporation can be provided as dynamic input, where the latter is used to estimate actual evaporation from lakes and reservoirs. In such case wind speed maps are not needed by the model." We have also clarified that "For this work, GLEAM was applied over the entire Po River Basin to produce both potential and actual evaporation estimates at 1 km resolution." The use of actual evaporation in Figure 3 has also been specified in the figure caption.

I would also like to hear the authors reasoning on why snow depth data and SM data was used in a data assimilation framework and not utilized in the calibration instead?

Reply:

Upon the reviewer's comment we have added In Sect 4.3: "It is worth noting that soil moisture and snow depth data were not used in this experiment because they are not model input variables but rather assimilation variables, hence the calibration procedure described in section 3.2 would not be directly applicable."

Figure 10: After reading the paper, I am still unsure what forcing data was used for the baseline run. Especially for the PET data I am quite unsure. I have a hard time understanding the last three experiments. For example PE is never defined and I am left guessing what the abbreviation means. If I understand the three last experiments correct, the model has been recalibrated, but why does the calibration performance not improve?

Reply:

To clarify the meaning of PE and Q, we have added in the caption of Figure 10 " In the three rightmost columns, PE stands for precipitation and evaporation, while Q stands for discharge." A table was inserted in the supplement material (Table S4) to clarify the datasets used in each model run. The calibration with satellite derived PE provided similar or slightly deteriorated performance in comparison with the original calibration because the high quality and resolution of in-situ hydro-meteorological data enable a more realistic search of the optimal parameter set in comparison to the use of satellite data, even when the model is then forced by the same satellite data used in calibration.

Specific comments

An additional fifth point to the benefits of EO data to enhance hydrological models:

Improved parametrizations (Many examples out there... such as: https://doi.org/10.5194/hess-22-1299-2018).

Reply:

We agree and have clarified that one key benefit is "3) as benchmark data for model calibration and improved parameterization". The suggested reference was added and cited in the Introduction section.

Page 3 line 85. Please explain what is meant by soil capacity.

Reply:

The sentence was improved and now reads: "for soil texture identification, we applied the USDA method (Shirazi and Boersma, 1984) using the ISRIC SoilGrids (Hengl et al., 2017) global maps of the fractions of sand and clay, combined with the ESA CCI SoilMoisture (Dorigo et al., 2017) global map of soil porosity."

The Continuum model is mentioned in section 2.1. and 2.2 prior to its introduction.

Reply:

The text has been changed to make it more consistent and omitting the model name before it is described in section 3.1, unless strictly necessary. In fact, this comment is partly in contrast with a comment by Reviewer #3 who asked for additional clarifications about why Continuum was used in this work at an early stage in the article, hence we have added in the introduction that Continuum is "CIMA's distributed hydrological model"

Reply to Referee #2

We thank the reviewer for his/her time in reading our manuscript and for the overall positive evaluation received. We do not disagree with any of the reviewer's comments so the vast majority of those have resulted in an addition to the text or to a change. Our reply to each comment is shown below, interspersed with the reviewer's comments.

1. While the results are presented in terms of statistical measures, the goodness in simulating high flows and low flows are obviously very different for different gauging stations. This is a major issue that needs attention and at least a discussion and some additional supplementary materials are required.

Reply:

We understand the reviewer's comment and agree that further work would help give more insight on the model behavior. We have added at the end of the Discussion section that "Given the number of experiments presented, focused on the role of different input data and model parameterization, results are only shown through overall statistics of each model run. Future work will investigate detailed model behavior over specific hydrological processes, regimes, seasonality and quantiles of the flow duration curve, to better disentangle strengths and weaknesses of the considered satellite products in specific hydrological conditions". In the work hereby presented, this would be outside the main focus, given the already large amount of information shown, and which may therefore distract the reader from the novelty aspects that we want to pass, which are:

   • We show recent advances in the development of five different high resolution satellite products.

   • Those satellite products are fed into a hydrological model (as forcing input and through data assimilation), first individually and then all together, and results show skillful results also compared to those of the same model driven by high quality ground observations.

   • We take early steps towards fully satellite-driven hydrological applications, by calibrating a hydrological model using satellite-based forcing input and optimizing the objective function using satellite-derived discharge estimates at five virtual stations as benchmark.

   • This work is part of the development of a Digital Twin Earth focused on the water cycle and hydrological processes, and contributes to the Destination Earth program launched by the European Commission.

2. L88-89, the ECOCLIMAP (2013) was used for the vegetation coverage. There are obviously more recent data for the period of the simulations. A comparison and quantification of the uncertainty is needed.

Reply:

Upon the reviewer's comment we have added in Sect. 2.1 "In the choice of spatial information, large scale datasets were deliberately used over more detailed local data, in line with the concept of the Digital Twin Earth and in view of the plan to extend the simulation area for a continental or global application". In addition, in our modeling experience, this dataset proved to work well and

gives additional information in comparison to alternative datasets, including stomatal resistance, mean canopy height, as well as over 200 vegetation classes, hence we preferred not to perform major changes to the static layers used, given the already wide range of model configurations tested in this work.

3. The MS needs to report the used signal to noise ratio that determines the weights in eq. (1-2).

Reply:

Upon the reviewer's comment we have added after Eq. 2 some details on the estimation of the signal to noise ratios used to calculate the weights. It reads: "SNR is estimated as the ratio between the variance of the true signal and that of the considered satellite product, multiplied by a parameter representing the systematic error (see Gruber et al. 2017), where the subscripts 1 and 2 refer to the SM2RAIN-ASCAT and IMERG-LR datasets, respectively". Also, following a comment by Reviewer #3, this part has been considerably enhanced with additional methodological details and an additional reference, which led to a more understandable text on how the satellite precipitation product was produced.

4. Datasets of different spatial resolution are mentioned, but what is the used spatial resolution in the hydrological modeling, 10km? How are they converted to the same resolution?

Reply:

We have slightly modified the sentence at the end of Sect. 3.1, which now reads: "For this work, S3M and Continuum were set up and run over the entire Po River basin (drainage area of 74,000 km$^2$), with a constant grid spacing of 1 km and time resolution of 1 hour." As mentioned in the manuscript, all dynamic input is generated at 1 km resolution except for the SM2RAIN precipitation. For the latter we have added in Sect. 1.2.1 the sentence "The 10 km resolution dataset thus generated was resampled at 1 km resolution through bilinear interpolation for use in the hydrological model."

5. L163-165: The used RT1 model for soil moisture retrieval uses auxiliary Leaf Area Index (LAI) time series provided by ECMWF ERA5-Land reanalysis dataset to correct vegetation effects, but ERA5- Land assumes a fixed land cover and static monthly leaf area index (LAI) climatology. How does the actual LAI change, in particular in agricultural areas, impacts the soil moisture and the subsequent use of it in the hydrological modeling?

Reply:

The scope of the article does not aim to investigate optimal soil moisture assimilation schemes but rather to use assimilation techniques and implementation strategies developed in previous research. Here, we agree with the concern raised by the reviewer and added a comment in the discussion section, which read "Our findings confirms the challenges in implementing a semi-automated assimilation of satellite soil moisture already pointed out in previous research (Laiolo et al., 2016; Wanders et al., 2014), where a range of factors affect and often decrease the assimilation performance, including the presence of complex topography, snow cover, frozen soil, urban areas, as well as differences between modeled and actual vegetation cover and leaf area index".

6. In eq. (4), G is assumed as 0.45, meaning the observation carries 45% information and the simulated background 55% information for all grids. This seems a gross simplification for different land cover types and needs at least a discussion and justification.

Reply:

The text in Sect. 3.3 was improved, with the addition of some relevant details. It now reads: "In this application we used a constant value of G=0.45, following the recommendations by Laiolo et al. (2016), who estimated optimal G values from a test on four different satellite-derived soil moisture products. In addition, we used G=0 in areas with low Pearson correlation coefficient (r<0.7) between satellite-derived and modeled soil moisture in the simulation period."

7. L314-316: 'Results denote a generally skillful reconstruction of river discharges for all experiments, with mean KGE at the 27 stations ranging between 0.13 (SM2RAIN+GLEAM) and 0.53 (C-SNOW), all well above the no-skill threshold of KGE0 = 1-2^1/2 $\cong$ -0.41 (see Knoben et al., 2019).' Please explain how 0.13 > 0.41?

Reply:

The no-skill threshold according to Knoben et al. (2019) is equal to -0.41 , being calculated as 1 minus the square root of 2, that is 1 - 1.41 $\cong$ -0.41. Perhaps in the article preprint this is slightly misleading because the minus sign is in line 315 while the 0.41 is in the following line due to page justification. We are confident that this will be solved in the final article version.

Reply to Referee #3

We thank the reviewer for his/her time in reading our manuscript and for the overall positive evaluation received. The Reviewer's evaluation was very comprehensive and comments were detailed, which benefited a lot the quality of the revised version. We do not disagree with any of the reviewer's comments so the vast majority of those have resulted in an addition to the text or to a change. Our reply to each comment is shown below, interspersed with the reviewer's comments.

Thank you very much for considering me as a reviewer for the manuscript by Alferi et al., titled "High resolution satellite products improve hydrological modeling in northern Italy" (hess-2021-632). I find the title of the study appropriate, thus read with much interest the manuscript written and presented in high quality. The study deals with the integration of multi-sensor and multi-resolution satellite products into hydrological modeling based on different experiments/simulations. Against the background of the mentioned "Digital Twin" of the Earth, this approach is of high relevance to the scientific community and implies the potential for future investigations. To my understanding, the study's introduction provides sufficient background and includes relevant references. About the description of the research design and methods applied, I see some need for improvement to further specify. According comments/questions are listed below. For the aspect of language and style, I do not see any serious flaws, thus only minor spell checking is recommended.

Reply:

We thank the reviewer very much for the overall positive evaluation of the manuscript. We have worked to improve the article following his/her recommendations and we believe it is now clearer and more readable. Replies to each point raised can be found in the following.

Line 14: Use the plural for resolution (i.e., spatial and temporal resolutions)

Reply:

Amended

Line 15: You write "high-resolution" here in this line, but in the title, you go for "high resolution". This is minor; however, I recommend using consistent writing.

Reply:

Thank you for pointing this out. All occurrences have been uniformed to "high resolution".

Line 16: As you have introduced EO as an acronym for Earth observation in line 13 already, I recommend using this acronym consistently.

Reply:

Amended as suggested.

Line 16: As the abstract acts as an "appetizer" to your study, I would prefer to read more specific information, thus recommend adding the number of experiments investigated (i.e., "In a set of six experiments, […]").

Reply:

Amended as suggested.

Lines 31 and 44: What is the order of your references listed? Alphabetical? Chronological?

Reply:

Following the journal guidelines, in-text citations can be ordered based on relevance, chronological or alphabetical listing, depending on the author's preference. We have opted for chronological order, with older references placed before and newer afterwards.

Line 66: No comma after Po River Basin.

Reply:

Amended.

Line 78: How was the spatial resolution of the DEM resampled from 90 m to 1 km spatial resolution?

Reply:

In the revised version we have clarified that "The DEM was upscaled at the chosen model resolution of 1 km through cubic resampling […]"

Line 80: What is the reasoning for an upstream area larger than 240 km$^2$? Did you choose?

Reply:

We have clarified in the text that "The river network is defined by cells with an upstream area larger than 240 km2, following previous applications of Continuum in northern Italy." In fact this threshold is very much related to the hydrological model used (Continuum) and its settings. Its physical meaning is related to a value where the effect of river routing along the river network becomes relevant in comparison to the effect of the slope runoff.

Line 81: What is the source of the high-resolution stream network of the main rivers?

Reply:

In the revised version we have specified that "To improve its spatial representation, the DEM was carved with a high resolution stream network of the main rivers taken from the Italian Institute for Environmental Protection and Research"

Line 83: The hydrological model Continuum used in the study is mentioned here for the first time. So far, the readership was not informed about the specifications of this model, thus reasoning why this model was chosen for the simulation experiments.

Reply:

Following the reviewer's comment we have pointed out in the first occurrence of Continuum (except for the Abstract) that "[...] we test the influence of five new high resolution satellite-derived datasets on the performance of CIMA's distributed hydrological model Continuum (Silvestro et al., 2013) set up for the entire Po River Basin in northern Italy"

Line 84: What do you mean specifically with a "hydrological soil type map"? What soil hydrological/hydraulic properties are provided?

Reply:

For consistency with the acronym of the database HYSOGs used we have rephrased it to "Hydrologic soil groups were extracted from the HYSOGs250m (Ross et al., 2018) [...]". We have also clarified that such information then contributes to the estimation of Curve Number maps: "The Curve Number map used to model direct runoff and infiltration from rainfall excess, was derived from the ESA-CCI 2018 Land Cover map (ESA, 2017) at 300 m resolution, together with information on the soil characteristics. [...]"

Line 85: What do you mean specifically with "soil capacity"?

Reply:

This part has been improved to be more specific and with a more correct terminology. It now reads "[...] for soil texture identification, we applied the USDA method (Shirazi and Boersma, 1984) using the ISRIC SoilGrids (Hengl et al., 2017) global maps of the fractions of sand and clay [...]"

Line 86: Use plural for "fraction", i.e., fractions of sand and clay.

Reply:

Amended

Line 87: So far, the datasets provided are represented by grids. I assume that the spatial data on the glacier areas is provided in vector data. If so, how was the data implemented?

Reply:

That is correct. Polygons of glacier areas were turned into a raster at the model resolution using the criterion of dominant class within each output cell. Being a standard approach we opted for omitting such a methodological detail, to allow the reader focusing more on the key datasets and research strategy.

Line 89: Your information on the vegetation coverage originates from the ECOCLIMAP dataset with 1 km spatial resolution. What is the reasoning for using this dataset whilst you have used the ESA CCI Land Cover product (300 m) for deriving your curve number?

Reply:

In our modeling experience, the ECOCLIMAP dataset proved to work well and gives additional information in comparison to the ESA dataset, including stomatal resistance, mean canopy height, as well as over 200 vegetation classes, hence for this work we opted for using a combination of the two land cover datasets.

Lines 92-95: You write of a set of variables relating to the dam reservoirs and natural lakes. How do those variables go into the parameterization of Continuum, particularly specific information on the weir length? This parameterization and the reasoning are not getting clear.

Reply:

Here we did not go into the details of how dams and lakes are modeled for multiple reasons. First because they do not play an active role in the different model experiments (no related model parameter is calibrated; dam/lakes parameters are not changed from one experiment to another). Second, no further analysis specific to their output is carried out, while the article is rather oriented to showing the influence of different dynamic input data, especially those derived from satellite data. Finally, we included in the "Code and data availability" section that "S3M and Continuum are open source models and their code is available at https://github.com/c-hydro ", so that all modeling features can be seen in details, favoring transparency and the reproducibility of experiments. In particular, as can be read in the related Github page, information on the weir length is used to model the maximum outflow that can be drained from a reservoir in high flow conditions, using weir flow equations.

Sub-chapter 2 "Data": Before reading extensive information on the various datasets, I would prefer to have information on the temporal domain (i.e., observation period) considered for your study. Moreover, I am missing information on the study area (area size, climate, physical properties, land cover) that would support the readability and interpretation of the further information provided and results.

Reply:

Upon the reviewer's comment we have added the observational period (2017-2019) at the end of the introduction section, to anticipate this information. Moreover, an entirely new section (~280 words) has been added (now Sect. 2.1 entitled "Case study – the Po River basin"), which gives several information on the case study and its key features for the purpose of hydrological modeling.

Line 110: What is the specific reasoning for selecting 22 stations for cal and five stations for val? What were your criteria for selection (position in the Po River Basin?, Data coverage? Data density?)?

Reply:

We thank the reviewer for this comment which helped us improve this part. We decided to move here a sentence initially placed in the discussion section, though not ideal in such place. Now it is clarified that "Validation stations were chosen to represent different areas of the Po basin, including a mix of small and large sub-catchments with varying influence of lakes and reservoirs."

Figure 1: Your figure looks very appealing. However, up to now, the meaning of the virtual stations (black dots) is not clear. Maybe you can add a general study workflow leading into this subchapter? Also, I am wondering if a smaller "overview map" (e.g., placed in the upper left corner of Fig. 1) would be meaningful.

Reply:

To avoid producing additional figures we decided to put all point information in Figure 1, including the location of virtual stations, even if described specifically in a later section. However, this is in compliance with the submission guidelines. Also, to optimize the use of space in the figure we favored latitude and longitude bars at the borders over an overview map.

Line 129: How were the two datasets rescaled? What is the common reference? What are the relative systematic differences between the products?

Reply:

Upon the reviewer's comment this part has been considerably enhanced with the following text (including an additional reference to Crow et al., 2015): "TC was applied to the triplet: SM2RAIN-ASCAT, IMERG-LR and the MCM radar-gauge precipitation dataset. Note that, unlike the use of random error variances as in Crow et al. (2015), weights calculated as in (2) do not require the assumption of null systematic differences between the datasets, thanks to the self-consistency of the signal-to-noise ratio (see Gruber et al., 2017 for further details). Before the weights can be used to merge the data sets, relative systematic differences (i.e., long term bias) have to be corrected to make weights obtained by (2) converge to the optimal weights in a least square sense (Crow et al. 2015). Given the nature of the precipitation signal (containing many null values) this rescaling has been done by means of a multiplicative factor to the mean with respect to MCM."

Line 132: Add the acronym "GPM" after "Global Precipitation Measurement" in line 117 to introduce this abbreviation.

Reply:

Added as suggested

Figure 2: When comparing Fig. 1 to Fig. 2, I recommend using the same "spatial extent" for both figures to increase the figures' readability.

Reply:

The two figures have almost the same spatial extent, which is centered on the Po River Basin. All maps have lat and lon labels, to identify locations univocally. As the reviewer may understand, it is rather challenging to coordinate at such a level of detail with a relatively large group of

contributors from different institutes. Hence unless the reviewer or the editor believe this slight difference is a major limitation we would prefer to keep the current version.

Lines 150 and 151: It is minor again, however, I recommend using the same style for writing the spatial resolutions (1 km vs. 1-km).

Reply:

Agreed. All instances were uniformed to the "1 km" version.

Figure 3: I recommend adding the boundary of your spatial domain into Fig. 3 a (left). If possible, I would prefer to see subfigure Fig. 3b (right) a bit enlarged to enhance readability.

Reply:

Figure 3 has been remade following the reviewer's suggestions.

Line 176: How is the "natural vegetation" composed? (e.g., woodland, grassland, etc.).

Reply:

We have added some details about the natural vegetation: "[...] with a median Pearson correlation of 0.55 for croplands and 0.65 over areas primarily covered by natural vegetation (i.e., tree, shrub, herbaceous cover)"

Line 177: Where is the Oltrepo station located? As you write of different stations many times, I am wondering if their positions can be indicated in Fig. 1?

Reply:

As a general choice ,stations used to validate the satellite products are kept separated from the others used in the hydrological model. Otherwise we should add to Figure 1 also snow and ET stations, though that would make the figure less readable. Hence, being only one station for soil moisture validation, we decided not to include a dedicated map, but rather to show its geographic coordinates (lon / lat) in the figure, under the title. In addition, upon the reviewer's comment, we have added the name of the municipality where the station is located: "Validation was performed using in situ soil moisture for the Oltrepo station (Bordoni et al., 2019) located in Canneto Pavese (PV, Italy) [...]"

Figure 4: In your caption, you mix between writing "full wording" (i.e., Surface Soil Moisture) and abbreviations (i.e., SM). I recommend choosing one style to make the caption(s) a more stand-alone version. For the shading indicating the SD, I found it very hard to "read" in both digital and color printed format. Maybe you can decrease the opacity a bit?

Reply:

We have improved the use of abbreviations in line with the reviewer's comment. I've double checked the figure and in the original version the shading is adequately visible. Perhaps it's an

issue with a lower resolution version for peer-review. Anyway, we thank the reviewer for pointing it out. We will make sure that the figure can be interpreted well in the final print layout.

Line 194: When you write that the observed snow data were processed, what do you mean specifically? How were they processed? How was their aggregation towards a daily resolution done (median or mean)?

Reply:

We have added some details on the processing, which now reads "Observed snow-depth data were processed by (1) setting to missing any negative value, (2) applying climatological thresholds for maximum and minimum snow depth to remove spikes, and (3) using a threshold on the 6-hour moving coefficient of variation to detect periods with grass interference (Avanzi et al., 2014). Data was then aggregated at daily resolution, and C-SNOW data were extracted for the same locations and data range. "

Figure 5: I would prefer reading measures of performance (R2, RMSE) in the scatterplot (Fig. 5d).

Reply:

RMSE and Pearson's correlation coefficient have been added to the scatter plot, as suggested.

Line 214: I would prefer seeing the position of the five stations indicated in a map to support orientation and interpretability. How did you deal with the fact of cloud coverage and its impact on the availability of the reflectance data?

Reply:

These are the virtual stations shown in Figure 1. This has now been clarified in the text. For conciseness, in the article we focus only on the main product features and its use in the described experiments, while specific questions and details can be addressed in the cited literature.

Figure 6: As you have provided the averaged NS and KGE in the paragraph, I would additionally prefer to see those station-based metrics in the according sub-figures. For the left column of Fig. 6 (a-e), a slight increase in the gaps between those sub-figures is recommended.

Reply:

The figure was remade following the reviewer's suggestions.

In line 226, your description of the methods applied starts. For the previous sub-chapters providing information on the datasets for integration, also methods and results are provided. I am wondering if this was part of your analysis too and part of other or previous studies. I find this part hard to follow and would therefore prefer having clarification. Also, since now the reader got a lot of information on your different EO satellite datasets, an overview table stating the spatial resolution, temporal resolution, and purpose (model parameterization, data assimilation) would

be helpful. As your spatial target model resolution is 1 km, more specific information on the resampling techniques from all different spatial resolutions (< 1 km and > 1 km) might be helpful.

Reply:

We are not sure we understood the reviewer's point correctly. However, all numerical results shown in this article, resulting from model runs or product validation is part of this study. Instead, everything produced previously (methods, ancillary information) is duly cited in the related literature.

Line 248: What are your performance measures to evaluate the "best match"? Maybe add in brackets.

Reply:

This information is described in more details a few lines below "The cost function, based on the Kling-Gupta Efficiency (Gupta et al., 2009), computes an error between the duration curves at each percentile, weighted with the logarithm of the upstream area, to give higher weight to the downstream stations without neglecting the contribution of the most upstream ones". We have also added a piece of text in the line commented by the reviewer to clarify the reference to the objective function described in the lines below: "We deployed a multi-site calibration procedure that iteratively searches the model parameterization that best matches the available discharge observations over the calibration period at the 22 considered calibration stations (Figure 1), through minimization of a cost function".

Line 250: I recommend adding the figure reference for Figure 1 in brackets after "[…] at the 22 considered calibration stations" to support orientation.

Reply:

Amended as suggested (see reply to comment above).

Lines 252-253 (and beyond): Personally, I prefer seeing those "feature symbols" (i.e., cf, ct, CN, ws) in italic letters.

Reply:

Amended as suggested, in line with submission guidelines. Mathematical symbols are now all typeset in italics.

Line 261: Specify "J".

Reply:

The sentence was modified to "The point that minimizes the cost function is used as the centre of the following iteration, until the algorithm converges to an optimal solution."

Line 264: Again, this is minor, but choose on consistent style (Sentinel 1 vs. Sentinel-1).

Reply:

All occurrences were turned to "Sentinel-1" as of the official naming on the European Space Agency website

Lines 270 and 272: Unfortunately, it is not getting very clear to me if G here refers to the same or two different variables (kernel function and gain). Please specify if needed.

Reply:

It is the same variable. In the revised version it has been clarified by keeping only one naming (Kernel function), while the term "gain" has been removed.

Figure 7: I would prefer knowing the locations of the basins and having more information available (e.g., land cover) to increase understanding of the interpretation.

Reply:

Labeling the 27 stations is not so straightforward. In addition to the 5 stations shown in Figure 7, the other 22 hydrographs are shown in the Supplement material. Hence we prefer avoiding such attempts and leave the connection between hydrographs and respective location through its name and the upstream area shown in the title of each panel.

Line 311: Specify towards potential evaporation (?).

Reply:

All experiments were run using (actual) evaporation, as required by the model, rather than potential evaporation. We have clarified that "For this work, GLEAM was applied over the entire Po River Basin to produce both potential and actual evaporation estimates at 1 km resolution." However, potential evaporation, when provided as input, is only used to estimate actual evaporation from lakes and reservoirs. This notion has been added in the text. The use of actual evaporation in Figure 3 has also been specified in the figure caption.

Figure 9: I very much like your figures in the entire manuscript. However, I am wondering if you could change the color code of the KGE in Figure 9 towards more purple (or else) colors to allow for better differentiation from the background (e.g., stream network).

Reply:

We have used a palette which follows recommendations for color blindness impairments (https://www.color-blindness.com/coblis-color-blindness-simulator/). Colors look interpretable and circles can be easily distinguished from the white background and from the river network (also thanks to a gray contour around each circle and from the fact that the river network is a line feature). Of course it has the limitations of a relatively small set of figure panels when printed on an A4 paper (yet, in the electronic version one can zoom in). Hence, we would prefer keeping this version if possible.

Figure 11: Please see my comment for Figure 9. Also, I would rather see a stand-alone version of Figure 11' caption instead of "like Figure 9".

Reply: See previous reply. The caption was modified as suggested.

Line 368: So far, the abbreviation PE (potential evaporation) was not introduced. Please do so in an adequate position.

Reply:

PE stands for precipitation and evaporation, either coming from conventional sources or from satellite products. In the revised version this has been clarified in the caption of Figure 10.

Figure 13: The legend placed in one of the subfigures (upper row, middle) is valid for all sub-figures, right? I would rather see it in a more meaningful position.

Reply:

Correct. Given the distribution of the scores obtained, the best placements (i.e., with enough space to make it readable) were in the top-middle and top-right position, hence we think that top-middle is a reasonable choice. It is probably quite subjective, though we don't see any clear disadvantage of such a choice.

Discussion: How is the river discharge affected by different land covers/land uses in your study area (e.g., upstream). I am missing a more critical discussion on this effect, as the effect of the land cover also on soil moisture should be more highlighted in terms of uncertainty (effect of vegetation, surface properties).

Reply:

This was not investigated in detail, because despite being of sure interest it is well covered by the existing literature and it is not among the research questions set out for this work. Given the already large number of analyses we decided not to include it in the article.

---

## Author Response (AR2)

Savona, 7/7/2022

MS No.: hess-2021-632
Title: High resolution satellite products improve hydrological modeling in northern Italy

Dear editor and editorial support team of HESS,
We would like to thank you and the reviewers for the positive evaluation of the article we presented. Please find below our reply to address the four comments raised by Reviewer#1. We hope that the article is now suitable for publication.

1) Thank you for proving more information on how the GLEAM data were used. Still, I think more information needs to provided. Originally, I understood that the aET was used as assimilation dataset. Now, I can read that aET can be used as dynamic input to the model. Usually, aET is simulated dynamically by a hydrological model. How does the model use aET as input? Where is the water taken from? I can also read that PET is still used to estimated aET over lakes and reservoirs. This is all very vague. Please specify... Also the baseline and the GLEAM scenario perform very similar with respect to KGE in runoff. Does that mean that the simulated aET in the baseline model is very similar to the aET from GLEAM or is aET not important for runoff predictions?

Reply:
When provided as input such as from GLEAM, AET and PET are used in the same way as in the original formulation described by Silvestro et al. (2013), depending on the water availability. See also reply to question 4). In particular, the evaporative water demand is first taken from the canopy interception and then from the soil layer.
In response to the last point of the question we have also added in Sect. 4.1 "Average evaporation in 2017-2019 computed by Continuum for the Po basin is 950 mm/year and results 21% smaller than the GLEAM average of 1200 mm/year in the same time span."

2) I would like to add to the authors second reply that snow depth and SM data may very well be used as calibration target. Correct, they are not input variables to a hydrological model, but we can use them as calibration targets and I am still wondering why the authors did not calibrate their model against all available variables and instead used them as assimilation variables? Here an example for using snow data in calibration: https://doi.org/10.1016/j.jhydrol.2021.126020 and also one example for SM data: https://doi.org/10.1029/2019WR026085

Reply:
We fully agree with the reviewer and did not mean in our previous response that snow depth and SM data could not be used for model calibration. Both assimilation and calibration can bring benefit to improving the model output. Parameter calibration is especially beneficial in the calibration period, while data assimilation enable model updating whenever new data is available, yet usually without changing the model parameters. In the experiment design of this research, snow depth data and SM data were chosen to be used as assimilation variables rather than for parameter calibration. We are confident that such choice does not decrease the scientific significance of our research.

3) There should be a discussion that puts some of the findings in perspective. For example, changing precip to SM2RAIN leads to a deterioration of KGE. Model parameters were obtained from a calibration against the radar/gauge precip dataset. IF the authors would calibrate against SM2RAIN the KGE results would likely be improved. The parameter set obtained from a calibration has to be used with caution when transferring it to another model, with e.g. different precip or PET inputs...

Reply:
We are surprised by this comment of the reviewer. We did perform the calibration suggested by the reviewer in his comment, using SM2RAIN as forcing input. That's actually one of the key activities and it is mentioned in various parts of the article, even in the abstract. To make this clearer, we have made some minor changes towards the end of the Introduction section, which now reads "Further, we take the first steps towards hydrological modelling fully relying on satellite data, by calibrating and subsequently running the model using SM2RAIN satellite precipitation and GLEAM evaporation as forcing, and satellite-based estimates of river discharge as benchmark data for the calibration". Interestingly, calibration against SM2RAIN did not improve KGE of the simulation forced by SM2RAIN. This is stated and then further commented in Sect. 4.3 "Results from simulation #2 forced by the same SM2RAIN and GLEAM used in the calibration shows the lowest performance among the three (mean *KGE*=0.07 over all 27 stations). Simulation #3 (EO PE, EO Q) gives satisfactory performance (mean *KGE*=0.10), relatively close to #1 (mean *KGE*=0.13), despite relying largely on satellite data.".

4) How is PET calculated in the baseline model. The authors mentioned the hourly weather variables which are likely used here. Please add more details.

Reply:

Upon the reviewer's comment we have added a reference to Silvestro et al. (2013) in Sect. 3.1, which describe in details the formulation used to compute the evaporation in the Continuum model. The new sentence reads "Evaporation is estimated through a bulk formulation by solving the mass and energy balance as described in Silvestro et al. (2013) and related appendix, though it can also be provided as input variable".